



# Modeling the influence of snowcover temperature and water content on wet snow avalanche runout

Cesar VERA VALERO[1], Nander WEVER[2], Marc CHRISTEN[1], and Perry BARTELT[1]

[1]WSL Institute for Snow and Avalanche Research SLF, Flüelastrasse 11, 7260 Davos Dorf, Switzerland.
[2]École Polytechnique Fédérale de Lausanne (EPFL), School of Architecture, Civil and Environmental Engineering, Lausanne, Switzerland.

*Correspondence to:* C. Vera (cesar.vera@slf.ch)

**Abstract.** Snow avalanche motion is strongly dependent on the temperature and water content of the snowcover. In this paper we use a snowcover model, driven by measured meteorological data, to set the initial and boundary conditions for wet snow avalanche calculations. The snowcover model provides estimates of snow depth, density, temperature and liquid water content. This information

is used to prescribe fracture heights and erosion depths for an avalanche dynamics model. We compare simulated runout distances with observed avalanche deposition fields using a contingency table analysis. Our analysis of the simulations reveals a large variability in predicted runout for tracks with flat terraces and gradual slope transitions to the runout zone. Reliable estimates of avalanche mass (height and density) in the release and erosion zones is identified to be more important than an

exact specification of temperature and water content. For wet snow avalanches, this implies that the layers where meltwater accumulates in the release zone must be identified accurately as this defines the height of the fracture slab and therefore the release mass. This is an interesting result because it indicates the critical role of fracture depth as an input parameter in avalanche simulations. Advanced thermomechanical models appear to be better suited than existing guideline procedure to simulate

wet snow avalanches when accurate snowcover information is available.

## 1   Introduction

Avalanche hazard mitigation has historically concentrated on catastrophic avalanches releasing from dry, high alpine snowcovers. There are many regions in the world, however, where wet snow avalanche problems are dominant. Increasingly, avalanche engineers require methods to consider

the avalanche hazard arising from frequent wet snow slides (Naaim et al., 2013).





The runout of wet snow avalanches is especially difficult to calculate because temperature and liquid water content (LWC) have a strong influence on the mechanical properties of snow (Denoth, 1982; Voytokskiy, 1977; Salm, 1982). There are two primary effects. Firstly, the compactive hardness of snow *decreases* with increasing water content (Salm, 1982) and secondly, the shear viscosity *decreases* with increasing temperature Voytokskiy (1977). When warm snow contains liquid water, the deformation mechanics is controlled by the liquid water content at the grain to grain contact, (Salm, 1982). Wet snow can be plastically deformed until it reaches "packed density". The low compactive strength of wet snow is revealed in granulometric investigations of avalanche deposits: wet snow granules are large, heavy and poorly sorted in comparison to granules in dry avalanches (Jomelli and Bertran, 2001; Bartelt and McArdell, 2009). Thus, the initial compaction of wet snow facilitates the formation of large, dense granules, leading to a significant increase in the *bulk* flow viscosity and cohesion of the avalanche (Bartelt et al., 2015). Another indication of the viscous and cohesive character of wet snow flows are the formation of levees with steep vertical shear planes in wet snow avalanche deposits (Bartelt et al., 2012b).

To model the increase in bulk flow viscosity of wet snow avalanches (that is, the lower flow velocities associated with wet snow flows), the Swiss guidelines on avalanche calculation recommend increasing the velocity squared turbulent friction (Salm et al., 1990). Wet snow avalanches are therefore treated as dense granular flows in the frictional flow regime (Voellmy, 1955; Bozhinskiy and Losev, 1998). Because measured velocity profiles of wet snow avalanches exhibit pronounced visco-plastic, plug-like character, they are often modeled with a Bingham-type flow rheology (Dent and Lang, 1983; Norem et al., 1987; Salm, 1993; Dent et al., 1998; Bartelt et al., 2005; Kern et al., 2009). Bartelt et al. (2015) uses cohesion to reduce the random kinetic energy of the avalanche core which effectively hinders avalanche fluidization and prevents the formation of mixed flowing/powder avalanches (Buser and Bartelt, 2015).

An increased bulk flow viscosity, however, is not the only mechanical change induced by warm, moist snow. The presence of liquid water on interacting snow surfaces *decreases* the magnitude of the *bulk* sliding friction coefficient. This decrease has been observed and quantified in many experiments, particularly those involving ski friction (Glenne, 1987; Colbeck, 1992). The decrease in sliding friction results in long-runout avalanches Naaim et al. (2013), making wet snow flows particularly dangerous.

The sensitivity of wet snow avalanche flow on temperature and moisture content makes predictions of avalanche runout difficult. For example, wet snow avalanches often occur after extreme precipitation events followed by intense warming. Because of differences in snowcover temperature and water content between the release and runout zones, wet snow avalanches can start in sub-zero temperatures and run into moist, isothermal snowcovers. That is, sub-zero release areas can lead to the formation of dry mixed flowing/powder type avalanches that transition at lower elevations to moist, wet flows. Clearly, a wet snow avalanche model must account for the initial temperature and





water content of the snowcover.

In this paper we use snowcover models to establish the initial and boundary conditions for wet
snow avalanche dynamics calculations. The primary goal is to investigate if better snowcover
temperature and water content predictions can improve the calculation of wet snow avalanche
runout. We specify snow cover information that is derived from detailed physics based snowcover
model simulations using **SNOWPACK** (Bartelt et al., 2002; Lehning et al., 2002). Avalanche
dynamics parameters will not be tuned, but are fixed within the framework of empirical functions
parameterized by density, temperature and moisture content (Vera et al., 2015, 2016). Our goal is
to obtain accurate runout and deposits area predictions without ad-hoc modifications to avalanche
model parameters. Instead of parameter optimization, we specify snow depth, density, temperature
and moisture content in both release (initial conditions) and entrainment zones (boundary condi-
tions) as input data for the model.


The approach consists of three basic steps (see Fig. 1):

1. Simulation of snowcover conditions using measured weather data as input

2. Simulation of avalanches using initial conditions defined by snowcover conditions

3. Contingency table analysis to define the statistical score of avalanche runout calculation

The procedure is applied to simulate twelve documented avalanche events, for which extensive
field measurements are available, including measurements from airborne laser-scans, drones and
photography and hand-held GPS devices. To determine how the procedure performs we compare
the area covered in the simulations with the deposit area measured in the field. Simulated runout
patterns are compared to field observations. The correspondence of observed deposits and calculated
deposits is checked using a dichotomous contingency table, splitting the terrain in four different
classes: hits, misses, false alarms and correct negatives.




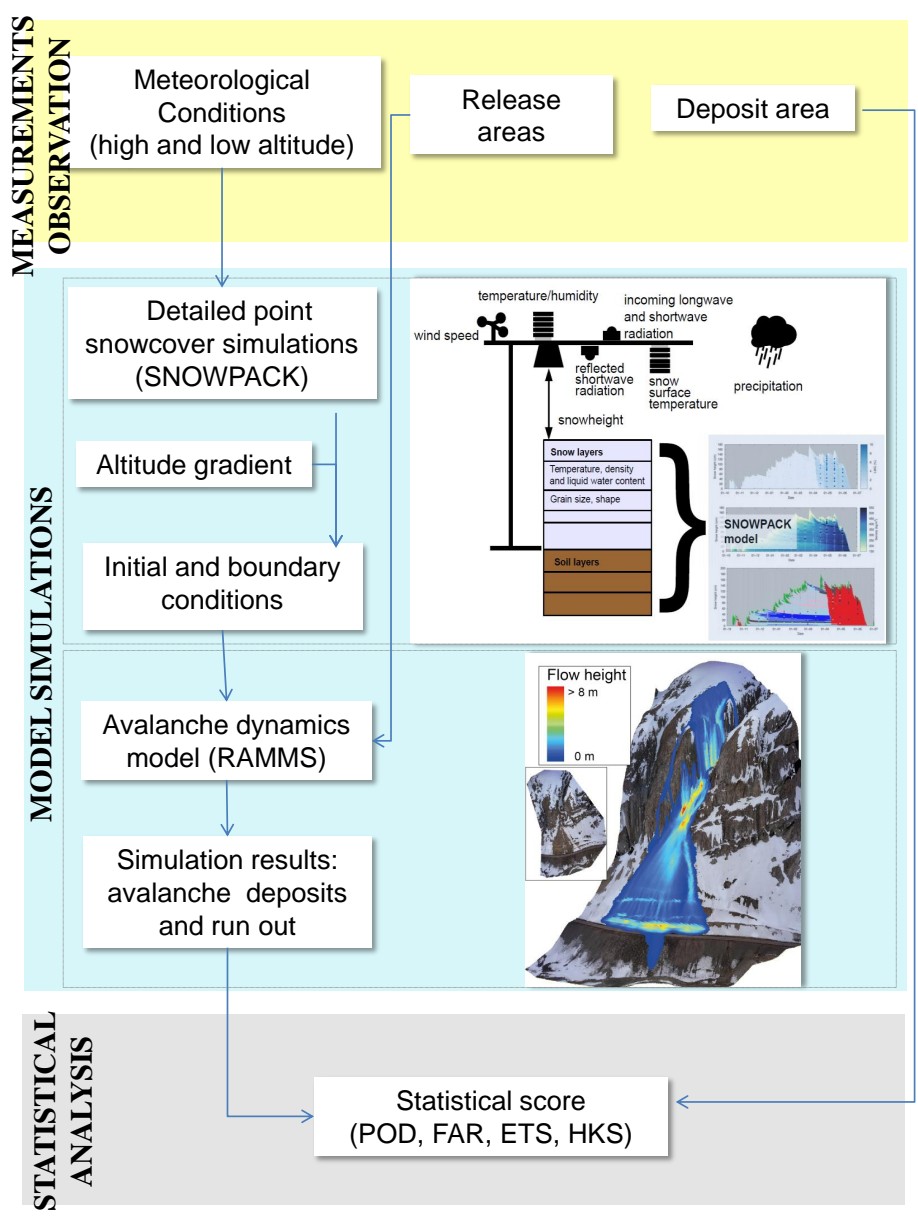

Fig. 1: Flow diagram depicting the three step model chain. The procedure begins by simulating snowcover conditions using measured weather data as input. Next, avalanche runout is simulated using initial and boundary conditions defined by snowpack modeling. Finally, a statistical score of the avalanche runout modeling is calculated.





Additionally, a sensitivity study is performed by interchanging the initial and boundary conditions of the twelve case studies and by varying the calculation grid cell size. The same contingency analy-

sis and runout comparison is performed with the results obtained from the sensitivity analysis. This establishes to what extend the initial and boundary conditions indeed control the model performance.

## 2  Wet snow avalanche modeling

Wet snow avalanche modeling necessitates the simulation of four physical processes (Vera et al.,

90  2015, 2016):

1. The rise in avalanche temperature by frictional dissipation.

2. Phase changes and the production of meltwater.

3. Entrainment of snow mass *and* the associated internal (thermal) energy change of the avalanche.

4. Constitutive models describing how the avalanche flow rheology changes as a function of temperature and moisture content.

One model that fulfills these requirements was developed by Vera et al. (2015, 2016). In this model, the flow of the dense avalanche core (subscript $\Phi$) is described by nine independent state variables:

$$\mathbf{U}_\Phi = (M_\Phi, M_\Phi u_\Phi, M_\Phi v_\Phi, R_\Phi h_\Phi, E_\Phi h_\Phi, h_\Phi, M_\Phi w_\Phi, N_K, M_w)^T. \tag{1}$$

These variables include the core mass $M_\Phi$ (which contains both the ice mass *and* the water mass $M_w$), the flow height $h_\Phi$, depth-averaged velocities parallel to the slope $\mathbf{u}_\Phi = (u_\Phi, v_\Phi)^T$ and in the slope perpendicular direction $w_\Phi$, the sum of the kinetic and potential energies associated with the configuration and random movement of snow particles $R_\Phi$ and the internal heat energy (temperature) $E_\Phi$.

The model equations can be written as a single vector equation:

$$\frac{\partial \mathbf{U}_\Phi}{\partial t} + \frac{\partial \boldsymbol{\Phi}_x}{\partial x} + \frac{\partial \boldsymbol{\Phi}_y}{\partial y} = \mathbf{G}_\Phi \tag{2}$$





where the components $(\mathbf{\Phi}_x, \mathbf{\Phi}_y, \mathbf{G}_\Phi)$ are:

$$\mathbf{\Phi}_x = \begin{pmatrix} M_\Phi u_\Phi \\ M_\Phi u_\Phi^2 + \frac{1}{2} M_\Phi g' h_\Phi \\ M_\Phi u_\Phi v_\Phi \\ R_\Phi h_\Phi u_\Phi \\ E_\Phi h_\Phi u_\Phi \\ h_\Phi u_\Phi \\ M_\Phi w_\Phi u_\Phi \\ N_K u_\Phi \\ M_w u_\Phi \end{pmatrix}, \ \mathbf{\Phi}_y = \begin{pmatrix} M_\Phi v_\Phi \\ M_\Phi u_\Phi v_\Phi \\ M_\Phi v_\Phi^2 + \frac{1}{2} M_\Phi g' h_\Phi \\ R_\Phi h_\Phi v_\Phi \\ E_\Phi h_\Phi v_\Phi \\ h_\Phi v_\Phi \\ M_\Phi w_\Phi v_\Phi \\ N_K v_\Phi \\ M_w v_\Phi \end{pmatrix}, \ \mathbf{G}_\Phi = \begin{pmatrix} \dot{M}_{\Sigma \to \Phi} \\ G_x - S_{\Phi x} \\ G_y - S_{\Phi y} \\ \dot{P}_\Phi \\ \dot{Q}_\Phi + \dot{Q}_{\Sigma \to \Phi} + \dot{Q}_w \\ w_\Phi \\ N_K \\ 2\gamma \dot{P}_\Phi - 2Nw_\Phi/h_\Phi \\ \dot{M}_{\Sigma \to w} + \dot{M}_w \end{pmatrix}.$$

$$(3)$$

The flowing avalanche is driven by the gravitational acceleration in the tangential directions $\mathbf{G} =$
$(G_x, G_y) = (M_\Phi g_x, M_\Phi g_y)$. The model equations are solved using the same numerical schemes
outlined in (Christen et al., 2010).

The model assumes non-zero slope perpendicular accelerations and therefore calculates the slope
perpendicular velocity of the core $w_\Phi$ (Buser and Bartelt, 2015; Bartelt et al., 2015). The center-
of-mass of the granular ensemble moves with the slope perpendicular velocity $w_\Phi$. When $w_\Phi >$
0, the granular ensemble is expanding; conversely when $w_\Phi < 0$, the volume is contracting. The
densest packing of granules defines the co-volume height $^0 h_\Phi^s$ and density $^0 \rho_\Phi^s$. The co-volume
has the property that $h_\Phi^s \geq^0 h_\Phi^s$ and $\rho_\Phi^s \leq^0 \rho_\Phi^s$. The normal pressure at the base of the column $N$ is
therefore no longer hydrostatic, but includes the impulsive reaction $N_K$ associated with the slope
perpendicular accelerations,

$$N_K = M_\Phi \dot{w}_\Phi. \tag{4}$$

The total acceleration in the slope perpendicular direction is denoted $g'$; it is composed of the slope
perpendicular component of gravity $g_z$, dispersive acceleration $\dot{w}_\Phi$ and centripetal accelerations $f_z$,
(Fischer et al., 2012). The total normal force at the base of the avalanche is given by $N$,

$$N = M_\Phi g' = M_\Phi g_z + N_K + M_\Phi f_z. \tag{5}$$

Changes in density are induced by shearing: Shearing in the avalanche core $\mathbf{S}_\Phi$ induces particle
trajectories that are no longer in line with the mean downslope velocities $\mathbf{u}_\Phi$ (Gubler, 1987; Bartelt
et al., 2006). The kinetic energy associated with the velocity fluctuations is denoted $R_\Phi^K$. The
basal boundary plays a prominent role because particle motions in the slope-perpendicular direction
are inhibited by the boundary and reflected back into the flow. The basal boundary converts the
production of random kinetic energy $R_\Phi^K$ into an energy flux that changes the $z$-location of particles
and therefore the potential energy and particle configuration within the core. The potential energy
of the configuration of the particle ensemble is denoted $R_\Phi^V$.



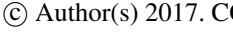
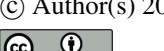

The production of free mechanical energy $\dot{P}_\Phi$, is given by an equation containing two model parameters: the production parameter $\alpha$ and the decay parameter $\beta$, see (Buser and Bartelt, 2009)

$$\dot{P}_\Phi = \alpha\left[\mathbf{S}_\Phi \cdot \mathbf{u}_\Phi\right] - \beta R_\Phi^K h_\Phi. \tag{6}$$

The production parameter $\alpha$ defines the generation of the total free mechanical energy from the shear work rate $[\mathbf{S}_\Phi \cdot \mathbf{u}_\Phi]$; the parameter $\beta$ defines the decrease of the kinetic part $R_\Phi^K$ by inelastic particle interactions. The energy flux associated with the configurational changes is denoted $\dot{P}_\Phi^V$ and given by

$$\dot{P}_\Phi^V = \zeta \dot{P}_\Phi. \tag{7}$$

The parameter $\zeta$ therefore determines the magnitude of the dilatation of the flow volume under a shearing action. When $\zeta = 0$ there is no volume expansion by shearing. For wet snow flows the value of $\zeta$ is small, $\zeta < 0.2$.

We model temperature dependent effects by tracking the depth-averaged avalanche temperature
$T_\Phi$ within the flow (Vera et al., 2015). The temperature $T_\Phi$ is related to the internal heat energy $E_\Phi$ by the specific heat capacity of snow $c_\Phi$

$$E_\Phi = \rho_\Phi c_\Phi T_\Phi. \tag{8}$$

The avalanche temperature is governed by (1) the initial temperature of the snow $T_0$, (2) dissipation of kinetic energy by shearing $\dot{Q}_\Phi$, as well as (3) thermal energy input from entrained snow $\dot{Q}_{\Sigma \to \Phi}$
and (4) latent heat effects from phase changes $\dot{Q}_w$ (meltwater production), see Vera et al. (2015). Dissipation is the part of the shear work not being converted into free mechanical energy in addition to the inelastic interactions between particles that is the decay of random kinetic energy, $R_\Phi^K$

$$\dot{Q}_\Phi = (1 - \alpha)\left[\mathbf{S}_\Phi \cdot \mathbf{u}_\Phi\right] + \beta R_\Phi^K h_\Phi. \tag{9}$$

A fundamental assumption of this model is that liquid water mass is bonded to the ice matrix of the
snow particles and therefore is transported with the flowing snow. Mathematically, the governing equations treat moisture content as a passive scalar. Meltwater production is considered as a constraint on the flow temperature of the avalanche: the mean flow temperature $T_\Phi$ can never exceed the melting temperature of ice $T_m = 273.15\ K$. The energy for the phase change is given by the latent heat $L$

$$\dot{Q}_w = L\dot{M}_w \tag{10}$$

under the thermal constraint such that within a time increment $\Delta t$

$$\int_0^{\Delta t} \dot{Q}_w dt = M_\Phi c(T_\Phi - T_m) \qquad \text{for} \qquad T > T_m. \tag{11}$$



Obviously, when the flow temperature of the avalanche does not exceed the melting temperature, no latent heat is produced, $\dot{Q}_w = 0$.

Another source of thermal energy is snow entrainment. The total mass that is entrained from the snowcover ($\Sigma$) is given by

$$\dot{M}_{\Sigma \to \Phi} = \rho_\Sigma \kappa \left\| \mathbf{u_\Phi} \right\|. \tag{12}$$

where $\rho_\Sigma$ is the density of snow and $\kappa$ the dimensionless erodibility coefficient. The liquid water mass entrained by the avalanche is therefore,

$\dot{M}_{\Sigma \to w} = \theta_\Sigma^w \dot{M}_{\Sigma \to \Phi}. \tag{13}$

where $\theta^w$ is the LWC of the entrained snow. The thermal energy entrained during the mass intake is

$$\dot{Q}_{\Sigma \to \Phi} = \left[ \theta_\Sigma^i c_i + \theta_\Sigma^w c_w + \theta_\Sigma^a c_a + \frac{1}{2} \frac{\left\| \mathbf{u_\Phi} \right\|^2}{T_\Sigma} \right] \dot{M}_{\Sigma \to \Phi} T_\Sigma \tag{14}$$

where $c_i$, $c_w$ and $c_a$ are the specific heat capacity of ice, water and air, respectively. When the snow layer contains water $\theta_\Sigma^w > 0$, then the temperature of the entire layer is set to $T_\Sigma = 0°$ C. Equation

14 takes into account the production of heat energy during the entrainment process.

To model frictional resistance $\mathbf{S}_\Phi = (S_{\Phi x}, S_{\Phi y})$ in wet snow avalanche flow we apply a modified Voellmy model(Voellmy, 1955; Salm et al., 1990; Salm, 1993; Christen et al., 2010),

$$\mathbf{S}_\Phi = \frac{\mathbf{u_\Phi}}{\left\| \mathbf{u_\Phi} \right\|} \left[ S_\mu + S_\xi \right]. \tag{15}$$

consisting of both a Coulomb friction $S_\mu$ (coefficient $\mu$) and a velocity dependent stress $S_\xi$ (coeffi-

cient $\xi$). The friction terms $S_\mu$ and $S_\xi$ are given by

$$S_\mu = \mu N - (1 - \mu) N_0 \exp \left( \frac{N}{N_0} \right) + (1 - \mu) N_0 \tag{16}$$

and

$$S_\xi = \rho_\Phi g \frac{\left\| \mathbf{u_\Phi} \right\|^2}{\xi}. \tag{17}$$

In the Coulomb friction term, $N_0$ is the cohesion; see Bartelt et al. (2015) for values of $N_0$ for wet

snow. To model the decrease in friction from meltwater lubrication, we make the Coulomb stress dependent on the meltwater water content $h_w$. We use the following lubrication function to replace the standard Coulomb friction coefficient $\mu$:

$$\mu(h_w) = \mu_w + (\mu_d - \mu_w) \exp \left[ -\frac{h_w}{h_s} \right]. \tag{18}$$

where $\mu_d$ is the dry Voellmy friction coefficient, $\mu_w$ is the limit value of lubricated friction (Voellmy

assumed this value to be $\mu_w = 0$ in the limiting case) and $h_s$ is a scaling factor describing the height





of the shear layer where meltwater is concentrated. The dry friction $\mu_d$ depends on the avalanche configuration:

$$\mu_d = \mu_0 \exp\left[ -\frac{R_\Phi^V}{R_0 + N_0} \right], \tag{19}$$

where $\mu_0$ is the dry Coulomb friction associated with the flow of the co-volume, which we take to be $\mu_0 = 0.55$, see (Buser and Bartelt, 2015). The parameter $R_0$ defines the activation energy for fluidization. Cohesion enhances the activation energy and therefore hinders the fluidization of the avalanche core (Bartelt et al., 2015).

### 3 Selected wet snow avalanche events and modeling procedure

We apply the numerical model to simulate documented wet snow avalanches. The data set includes twelve wet snow avalanches that occurred in the Swiss Alps and in the Chilean central Andes between 2008 and 2015. The avalanches were selected for three reasons: (1) the avalanche was located in the vicinity of an automatic weather station (henceforth AWS), (2) the release area and the area inundated by the avalanche were measured either by hand held GPS, drone or terrestrial laser scanning and (3) a high resolution digital elevation model (i.e. 2 m or higher) is available to simulate the terrain. This information is summarized in Table 1. The avalanche release volumes varied between 7,000 m$^3$ and 330,000 m$^3$. Most avalanches released from a wet snowcover and entrained additional wet snow. However, in three events (Grengiols, Braemabuhl Verbauung and Gatschiefer) the avalanche released as a dry slab at subzero temperatures, but entrained warm, moist snow at lower elevations. The release, transit and deposit zone of ten of the twelve case studies were additional photographed from a helicopter. The two remaining avalanches (Drusatscha and Braemabuhl 2013) were photographed by the authors from the deposition zone. The measurements from the release areas and deposits outlines for every avalanche path are shown in Supplement A in the online supplement.

### 3.1 SNOWPACK simulations

The data provided by the automatic weather stations allows us to run detailed, physics based snowcover simulations. We apply the **SNOWPACK** model (Bartelt et al., 2002; Lehning et al., 2002; Wever et al., 2014) in a similar setup as the snow-height driven simulations in Wever et al. (2015, 2016). Because **SNOWPACK** is a one-dimensional model, we must transfer point simulation results to the slope in order to apply a three-dimensional avalanche dynamics model. The horizontal distance between release zone and meteorological station varied between 200 m (the nearest) and 2200 m (the farthest). More important than the linear distance is the difference in altitude. The small elevation difference between the release zones and the weather stations, (see Table 1), provides the sufficient conditions to apply snowcover models to estimate the initial and boundary conditions of the case studies (Vera et al., 2016; Wever et al., 2016).




Table 1: Case study, date and estimated time of occurrence, (AWS) automatic weather station and virtual slope used at the top and at valley bottom for the release zone and deposits area, type of field measurement and altitude of the release and of the deposits in m.a.s.l.

| Avalanche | Date/Hour | AWS-slope Top/Valley (altitude AWS m.) | Measurements | Altitude release/Deposits (m) |
|---|---|---|---|---|
| Gruenbodeli | 23.04.2008 ≈ 14h00m | KLO2-NE (2140)/ SLF2 (1550) | Laser scan | 1900/1600 |
| Salezer | 23.04.2008 15h00m | WFJ2-W (2560)/ SLF2 (1550) | Laser scan | 2400/1500 |
| Gastschiefer | 23.04.2008 16h00m | KLO3-N (2310)/ SLF2 (1550) | Laser scan | 2400/1200 |
| Braemabuhl 2013 | 18.04.2013 15h00m | WFJ2-NE (2560)/ SLF2 (1550) | GPS profile | 2200/1600 |
| Drusatcha | 15.04.2013 17h00m | WFJ2-W (2560)/ SLF2 (1550) | GPS profile | 2200/1700 |
| MO-4 Andina Chile | 15.10.2013 19h15m | CAND5-SE (3520)/ Lagunitas (2770) | Ortophoto | 3700/3200 |
| Grengiols | 26.12.2013 13h00m | GOMS-NE (2450)/ Estimated | GPS profile | 2300/1400 |
| Verbier Mont Rogneux | 13.03.2014 17h00m | ATT2-W (2545)/ Estimated | GPS profile | 2400/1700 |
| Verbier Ba Comb | 13.03.2014 17h00m | ATT2-SW (2545)/ Estimated | GPS profile | 2200/1600 |
| Braemabuhl verbauung | 03.04.2015 12h00m | WFJ2-NE (2560)/ SLF2 (1550) | GPS profile | 2200/1600 |
| Braemabuhl Wildi | 04.04.2015 ≈ 14h00m | WFJ2-NE (2560)/ SLF2 (1550) | Drone photogrametry | 2200/1600 |
| CV-1 Andina Chile | 19.10.2015 17h00m | CAND5-E (3520)/ Lagunitas (2770) | Drone photogrametry | 2700/2500 |

To determine the initial temperature and moisture content of the snowcover requires an accurate modeling of the surface energy fluxes (sensible and latent heat exchanges, incoming short and long-wave radiation) which are influenced by the slope exposition. We account for exposition effects on surface energy fluxes using the virtual slope concept proposed by Lehning et al. (2008), which was found to provide accurate slope simulations that correspond with wet snow avalanche activity,

(Wever et al., 2016; Vera et al., 2016). We obtain snowcover layering, temperature, density and LWC in the release zones using virtual slope angles of 35° (see Table 2). The real slope angles of the release zones varied between 32° and 45°. Shortwave radiation measured at the AWS as well as snowfall amounts are re-projected onto these slopes, taking into account the exposition of the slope, (Lehning et al., 2008).

To model the snowcover at lower elevations in the transit and runout zones, we use meteorological data measured at the valley bottom. This information provides us with the snow temperature, snow height, density and LWC at lower elevations. In eight of the twelve case studies, the snowcover in the avalanche model can be considered as a single homogeneous layer while for the remaining case studies, the snowcover was best modeled as a two layer system consisting of old wet snow

covered by dry new snow, see Table 3. The elevation dependent properties of the snowcover along the avalanche path were determined by constructing a linear gradient between the upper and lower meteorological stations. This procedure could be applied for the case studies that occurred near Davos (seven case studies) and the cases in Chile (two cases).

     For the remaining case studies (Verbier Mont Rogneux, Verbier Ba Combe and Grengiols) we

estimated snowcover conditions along the avalanche track by applying a negative linear gradient of one third of snowcover height per 1000 meters of altitude. This rule provides gradients of snowcover depth of 2 cm to 6 cm per 100 meters of elevation (see Table 3). This method is in agreement with





Table 2: Initial conditions derived from SNOWPACK simulations at the release for each avalanche

| Avalanche | Date | Meteostation | LWC (%) | depth (m) | density (kg m$^{-3}$) | temperature (°C) | Cohesion (Pa) | Released Volume (m$^3$) | Growth index (-) |
|---|---|---|---|---|---|---|---|---|---|
| Gruenbodeli | 23.04.2008 ≈ 14h00m | KLO3-NE | 1.45 | 0.56 | 197 | -0.3 | 100.0 | 52882 | 2.2 |
| Salezer | 23.04.2008 ≈ 15h00m | ATT2-SW | 1.89 | 0.95 | 317 | -0.1 | 150.0 | 46394 | 2.4 |
| Gatschiefer | 23.04.2008 16h00m | KLO3-N | 1.63 | 1.72 | 320 | -0.1 | 150.0 | 330544 | 1.8 |
| Braemabuhl 2013 | 18.04.2013 15h00m | WFJ2-NE | 2.97 | 1.11 | 353 | 0.0 | 150.0 | 21404 | 3.5 |
| Drusatscha | 15.04.2013 17h00m | WFJ2-W | 3.41 | 0.54 | 291 | 0.0 | 150.0 | 32730 | 2.3 |
| MO-4 Andina Chile | 15.10.2013 19h15m | CAND5-SE | 2.44 | 0.90 | 296 | -0.2 | 150.0 | 9257 | 2.1 |
| Grengiols | 26.12.2013 ≈ 13h00m | GOMS-NE | 0.00 | 1.10 | 175 | -7.4 | 100.0 | 129392 | 3.9 |
| Verbier Mont Rogneux | 13.03.2014 17h00m | ATT2-W | 3.67 | 0.60 | 317 | 0.0 | 150.0 | 55817 | 1.8 |
| Verbier Ba Combe | 13.03.2014 17h00m | ATT2-SW | 3.40 | 0.58 | 349 | 0.0 | 150.0 | 21349 | 2.1 |
| Braemabuhl verbauung | 03.04.2015 12h00m | WFJ2-NE | 1.01 | 1.10 | 285 | 0.0 | 150.0 | 6858 | 2.7 |
| Braemabuhl Wildi | 04.04.2015 ≈ 14h00m | WFJ2-NE | 1.23 | 1.10 | 245 | -1.4 | 100.0 | 45614 | 3.3 |
| CV-1 Andina Chile | 19.10.2015 17h00m | CAND5-E | 2.36 | 0.95 | 359 | -0.1 | 150.0 | 4019 | 2.2 |

Table 3: Erosion conditions derived from the snowcover simulations for each avalanche case study. Upper and lower denotes two different erosion layers. The two layers system was used when new snow was lying over old snow cover and both layers were part of the studied avalanche. In case of only one layer all the fields at the second layer lower layer are set to zero.

| Avalanche | LWC (%) | | Erosion depth (m) | | Erosion depth gradient (m/100m) | | density (kg/m$^3$) | | volwater (mm/m) | | temperature (°C) | | temperature gradient (°C/100m) | | erodibility (-) | |
|---|---|---|---|---|---|---|---|---|---|---|---|---|---|---|---|---|
| | upper | lower | upper | lower | upper | lower | upper | lower | upper | lower | upper | lower | upper | lower | upper | lower |
| Gruenbodeli | 1.45 | - | 0.56 | 0.00 | 0.02 | - | 197 | - | 8.1 | - | -0.2 | - | 0.0 | - | 0.8 | - |
| Salezer | 1.89 | - | 0.95 | 0.00 | 0.03 | - | 317 | - | 18.0 | - | 0.0 | - | 0.0 | - | 0.7 | - |
| Gatschiefer | 0.00 | 1.47 | 0.55 | 0.95 | 0.03 | 0.04 | 185 | 360 | 0.0 | 14.0 | -1.0 | 0.0 | 0.0 | 0.0 | 0.6 | 0.7 |
| Braemabuhl 2013 | 2.97 | - | 1.11 | 0.00 | 0.04 | - | 353 | - | 33.0 | - | 0.0 | - | 0.0 | - | 0.6 | - |
| Drusatscha | 3.41 | - | 0.54 | 0.00 | 0.02 | - | 291 | - | 18.4 | - | 0.0 | - | 0.0 | - | 0.6 | - |
| MO-4 Andina Chile | 2.44 | - | 0.90 | 0.00 | 0.03 | - | 296 | - | 22.0 | - | 0.0 | - | 0.0 | - | 0.6 | - |
| Grengiols | 0.00 | 4.67 | 0.43 | 0.60 | 0.03 | 0.00 | 175 | 270 | 0.0 | 28.0 | -7.4 | 0.0 | 1.5 | 0.0 | 0.7 | 0.8 |
| Verbier Mont Rogneux | 3.00 | - | 0.60 | 0.00 | 0.02 | - | 317 | - | 18.0 | - | 0.0 | - | 0.0 | - | 0.6 | - |
| Verbier Ba Combe | 2.59 | - | 0.58 | 0.00 | 0.02 | - | 349 | - | 15.0 | - | 0.0 | - | 0.0 | - | 0.6 | - |
| Braemabuhl verbauung | 0.00 | 1.41 | 0.25 | 0.85 | 0.00 | 0.04 | 158 | 335 | 0.0 | 12.0 | -2.0 | 0.0 | 0.0 | 0.0 | 0.8 | 0.8 |
| Braemabuhl Wildi | 0.00 | 1.25 | 0.30 | 0.80 | 0.00 | 0.03 | 164 | 335 | 0.0 | 10.0 | -2.0 | 0.0 | 0.0 | 0.0 | 0.6 | 0.6 |
| CV-1 Andina Chile | 1.51 | - | 0.37 | 0.00 | 0.00 | - | 359 | - | 5.6 | - | -0.1 | - | 0.0 | - | 0.6 | - |

the Swiss Hydrological atlas. In these special cases, the snow temperature, density and LWC were kept constant to the values estimated by the **SNOWPACK** model at the release altitude. In case

of avalanches with new snow on top of the wet old snowcover, we consider the new snow amount measured at the AWS and estimate a decreasing linear gradient of new snow depth with altitude.

### 3.2   Avalanche dynamics calculations: initial and boundary conditions

We apply two different models to simulate the twelve case studies. The first is based on the

thermomechanical avalanche dynamics equations presented in Section 2, see (Vera et al., 2015, 2016); the second avalanche model follows the Swiss guidelines on avalanche calculation (Salm et al., 1990; Christen et al., 2010). The numerical model is outlined in Gruber and Bartelt (2007). Both models are implemented in the **RAMMS** software. Models and model parameters are compared in Table 4.


In the calculations, we are primarily concerned with the initial and boundary conditions, which



Table 4: Overview of model and model parameters used to simulate the twelve case studies.

| | VS guidelines | Thermomechanical | Comments |
|---|---|---|---|
| Reference | Salm et al. (1990) | Vera et al. (2015, 2016) | Both models in **RAMMS** |
| | Gruber and Bartelt (2007) | Buser and Bartelt (2015) | Christen et al. (2010) |
| $\mu_0$ (−) | Calibrated/guidelines | 0.55 | Reduced by lubrication |
| $\mu_w$ (−) | None | 0.12 | Constant in all simulations |
| $\xi_0$ (m s$^{-2}$) | Calibrated/guidelines | 1300 | Reduced by fluidization |
| $N_0$ (Pa) | 200 | 200 | Measured, see Bartelt et al. (2015) |
| $\alpha$ (−) | 0.00 | 0.05 - 0.07 | Depends on roughness |
| $\beta$ (1/s) | None | 1.0 | Depends on temperature |
| $R_0$ (kJ/m$^3$) | None | 2 | Constant in all simulations |
| $h_m$ (m) | None | 0.1 | Size of lubricated layer |
| $\kappa$ (−) | None | 0.6 - 0.8 | VS guidelines no entrainment |

are given by the snowcover model simulations; the release area is given by the field measurements. The fracture depth is defined by the location of the highest water accumulation within the snowcover (Wever et al., 2016) as was previously suggested by (Vera et al., 2016). Once the fracture depth

is known we set the snow density, snow temperature and liquid water values as the mean values over the slab which extends from the location of the maximum liquid water to the snow surface. We take the values at the estimated time of avalanche release. These values are shown in Tables 2 and 3. The amount of erodible snow along the path is estimated calculating a gradient between the snowcover conditions at the release and the conditions at the valley bottom. The erosion model used

is described by Christen et al. (2010); Bartelt et al. (2012a).

Once the initial and boundary conditions were found, the first set of simulations using the extended model were performed. As input parameters, the model uses the release area (measured), the snowcover initial conditions (calculated) and a set of friction and avalanche parameters. The

avalanche parameters were found by Buser and Bartelt (2009); Vera et al. (2015); Buser and Bartelt (2015). These parameters were kept constant for all 12 case studies as in (Vera et al., 2016). The fluidization parameter $\alpha$ (see Bartelt et al. (2006); Vera et al. (2016)), was fixed to a pre-determined value based on the terrain characteristics for each avalanche path. Once this parameter was fixed it was not tuned for the remaining set of simulations.

To perform standard Voellmy-Salm snow avalanche simulations following the Swiss guidelines (Salm et al., 1990) it is necessary to include the entire avalanche mass within the release volume. The guidelines do not consider entrainment along the avalanche path and therefore erosion was not considered in the Voellmy-Salm simulations. This procedure was adopted to follow as closely as possible the Swiss guideline procedures for avalanche calculations and allows a comparison

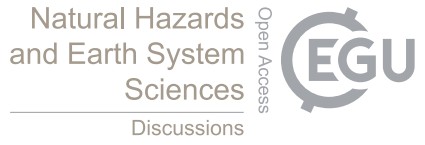



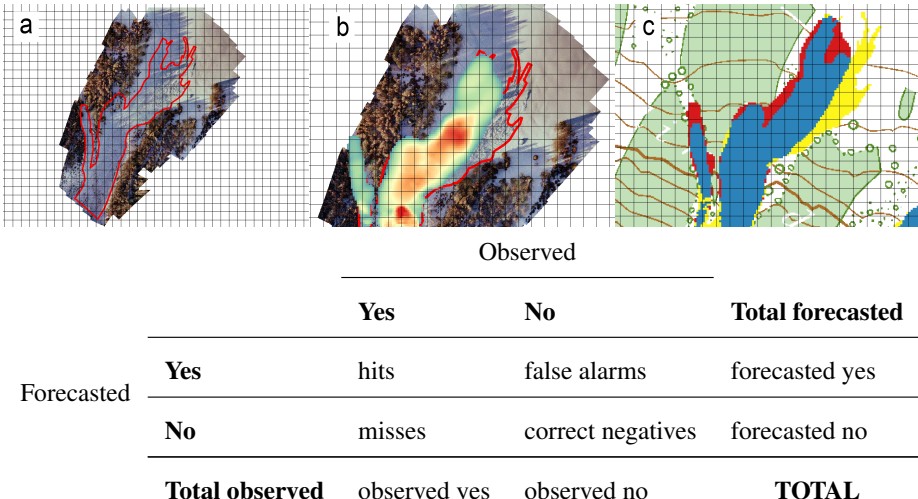

|  | Observed | | |
| --- | --- | --- | --- |
|  | **Yes** | **No** | **Total forecasted** |
| Forecasted **Yes** | hits | false alarms | forecasted yes |
| **No** | misses | correct negatives | forecasted no |
| **Total observed** | observed yes | observed no | **TOTAL** |

Fig. 2: Method to construct the contingency table, based on measured deposits outline (a), which is then combined with the simulated deposits area (b) to identify hits (blue), false alarm (red), misses (yellow) and correct negatives (no color, map only) (c).

between models which consider entrainment conditions (extended model) and models which employ calibrated parameters (Voellmy-Salm). The avalanche mass of the release area was estimated from the final mass (released plus eroded) calculated using the extended model. The total mass calculated in the extended model is concentrated in the measured release area. With this approach, a higher release depth is obtained, in comparison to model calculations with entrainment. This method

ensures that the total mass in both simulations is similar. The Swiss guidelines provides the user a set of friction parameters to use depending on the avalanche size and avalanche return period. Those friction parameters correspond to extreme, fast moving, dry-flowing avalanches which have longer runouts than wet ones. For the 12 case studies, the friction parameters used are the ones corresponding to the class 'Small' avalanches and return period of 10 or 30 years. This parameter

combination led to the overall best fit to observations. The calculations were performed with the same terrain and grid resolution.

### 3.3 Contingency table analysis for deposition area

The results obtained with the two models are compared through a statistical contingency table

analysis. We compare the area covered by the avalanche deposits calculated with both models with the deposits area measured for each case study. The terrain is divided in squared cells which correspond with the calculation cells used in the avalanche simulations (see Fig. 2 (a) and (b)).



| | |
|---|---|
| $\text{FAR} = \dfrac{\text{false alarms}}{\text{hits} + \text{false alarms}}$ | $\text{POD} = \dfrac{\text{hits}}{\text{hits} + \text{misses}}$ |
| $\text{HKS} = \dfrac{\text{hits}}{\text{hits} + \text{misses}} - \dfrac{\text{false alarms}}{\text{false alarms} + \text{correct negatives}}$ | $\text{ETS} = \dfrac{\text{hits} - \text{hits}_{random}}{\text{hits} + \text{misses} + \text{false alarms} - \text{hits}_{random}}$ [1] |

[1] where

$$\text{hits}_{random} = \frac{(\text{hits} + \text{misses})(\text{hits} + \text{false alarms})}{\text{total}}$$

Table 5: Mathematical definition of the statistics scores: probability of detection (POD), false alarm rate (FAR), Equitable threat score (ETS) and Hanssen Kuijpers or true skill score skill score (HKS)

For each cell we check whether the cell was covered by the observed avalanche deposits or not and whether the cell was covered by the avalanche simulation once the simulation stops or not. A
cell will be considered as covered by the avalanche simulations only if the calculated flow height with the mass at rest is more than 20 cm corresponding approximately to two granules diameter (Bartelt and McArdell, 2009). The correspondence of observed deposits and calculated deposits is checked using a dichotomous contingency table (see Table 2), that split the terrain in four different classes: hits, misses, false alarm and correct negatives (see Fig. 2(c)). Computing the amount of
cells for each class allows to calculate different metrics to judge how both models perform. In this study the probability of detection (POD), false alarm rate (FAR), equitable thread score (ETS) and Hanssen-Kuipers skill score or true statistic score (HKS) (see table 5) are calculated (Woodcock, 1976). For POD, ETS and HKS a score of 1 would mean a perfect score, in the case of FAR a score of 0 would indicate the perfect score.


### 3.4 Avalanche runout

In addition to the contingency analysis study for the inundated area, runout distance are analyzed. The runout distance was calculated from the difference in meters between the maximum distance reached by the avalanche in the measurements and the avalanche simulation calculated over the line
of steepest descend for each avalanche path in a DEM smoothed to a resolution of 20 m (see Fig. 3). The line of steepest descend was chosen as the longest line of steepest descend among all the possible ones departing from the depicted release area for each avalanche path.

### 3.5 Influence of initial conditions on avalanche runout: sensitivity study

To investigate how initial conditions influenced the avalanche runout and area covered by the
deposits, we performed 432 simulations on the twelve avalanche tracks where we interchanged the initial and boundary conditions from the 12 different initial and boundary conditions: from each of the twelve case studies we performed three different sets of simulations (3x12x12). As a sensitivity



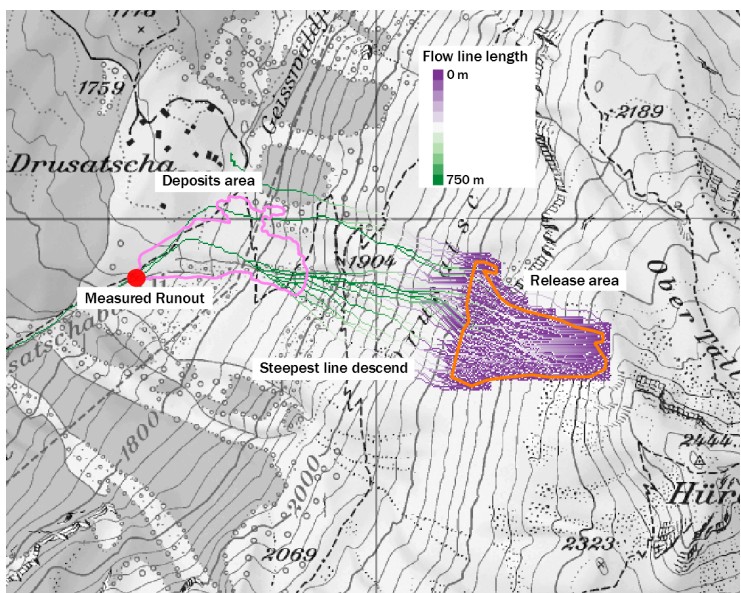

Fig. 3: Run-out distance calculation procedure. From each calculation cell at the release area the line of steepest descend is calculated. The intersection of the lowest part of the avalanche deposits with the longest calculated (red dot) define the avalanche runout. The same procedure is repeated with the simulation results. The distance measured on the steepest line between the the two intersection points is defined as the runout calculation error.

analysis we determined the difference between the observed and simulated runout as a function of the initial and eroded temperature, initial moisture content, fracture depth and snow density.


The sensitivity of the model to changes in the snowcover conditions was additionally evaluated. For this purpose, the same contingency analysis was performed for three different simulation sets constructed by varying the initial and boundary conditions for each avalanche path used in this study.

The three sets of simulations were constructed as follows:

1.  Twelve simulations for each avalanche path interchanging the initial and boundary conditions (fracture and erosion depth, snow temperature, density and LWC at the erosion and at the release) for the twelve different avalanches, obtaining thereby a set of 144 simulations.

2.  A second set of simulations was performed by using the snow temperature and LWC that was simulated by the snowcover model for that track. However, we varied the release and erosion





depths and the snow density of the twelve different case studies. This set contains another 144 simulations and is used to verify the model sensibility to changes in avalanche mass at the release and at the erosion.


3. A third set of simulations is constructed by keeping the snow depths and snow densities constant. The remaining conditions (i.e., temperature and LWC) were taken from the twelve case studies, leading to another set of 144 simulations, to investigate the importance of snowcover properties in relation to snowpack mass.


In total 432 simulations were performed for the entire sensitivity analysis, thirty six for each of the twelve avalanche paths.

## 4   Results

The contingency table analysis is used to explore the following questions:

1. Is it possible to drive avalanche dynamics calculations with initial and boundary conditions derived from snowcover modeling? Does the application of thermomechanical models improve the area covered by avalanche deposits and runout distances?

2. How sensitive are the simulated deposit areas and runout distances to released mass and snowcover properties?

3. What role does the calculation grid resolution play in the simulated areas covered by the deposits and runout distances?

### 4.1   Comparison between the guideline-VS and thermomechanical model

The twelve avalanche events were simulated using the guideline-VS model (Salm et al., 1990) and the thermomechanical wet snow avalanche model presented in Section 2. Recall that the guideline
friction parameters were used for wet snow avalanches and best overall fit to the observed inundation areas was found using the classification small and frequent return period of 10 to 30 years. The thermomechanical model used the fracture and entrainment depths derived from the snowcover modeling. Bulk snow temperature and moisture contents were determined by layer averaging of the fracture depth. The contingency table analysis for deposition areas and runout distances are shown
in Fig. 4.

A comparison between the guideline-VS and the wet snow avalanche model reveals that the thermomechanical model obtains significantly better results than the guideline-VS model The probability of detection (POD) in conjunction with false alarm rate (FAR) scores achieved by the





thermomechanical model improve the results by more than 0.15 points (see Fig. 4). The equitable
threat score (ETS) achieved by the thermomechanical model improves the guideline procedure
by more than 0.1 points (see Fig. 4). Additionally, the Hanssen and Kuipers or true skill score
(HKS) reached by the thermomechanical model improves by 0.19 points in comparison to the HKS
reached by the guideline model. Therefore, the thermomechanical model statistically outperforms
the guideline procedure in all four contingency metrics. The fact that the difference in ETS score
between the thermomechanical model and guideline procedure is higher than the difference in HKS
score shows that the HKS score is weighted toward detection, and thus POD, when the area covered
by the deposit of an avalanche is small compared to calculation domain (i.e., hitting pixels with
the avalanche deposits becomes a rare event). In contrast, the ETS penalizes both misses and false
alarms and therefore, guideline simulations which overran the measured deposit area (see in the on-
line Supplement) have increased FAR, and a stronger reduction in ETS scores in comparison to HKS.

The difference in performance between guideline-VS and thermomechanical wet snow avalanche
model simulations differ per avalanche path (see Fig. 4). The guideline-VS procedure has particular
difficulties with tracks containing a smooth transition between the acceleration and deposition zones.
These avalanche paths have a long distance where the steepness is getting progressively flatter (i.e.
Braemabuhl, Mont Rogneux, Ba Combe and Drusatcha, see in the online Supplement). In contrast,
the guideline-VS model does much better on avalanche paths with a sharp transition between the
acceleration and runout zones (Gruenbodeli, Salezer and Gatschiefer). In the examples where the
slope angle changes smoothly the guideline calculations systematically overran the measured de-
posits (Braemabuhl, Wildi, Mont Rogneux, Ba Combe). Thus, the guideline-VS does achieve good
scores on detection (POD) but is at the same time exhibiting a high false alarm rate (FAR).

The thermomechanical model performs equally well on both types of slope and is able to
reproduce runout distances on slopes with gradual transition to the runout zone. In the case of
Grengiols, the runout distance is somewhat underestimated; however, this was found to be caused
by the uncertainty of the elevation of the snowfall limit. This is an important result since it indicates
that the snowcover modeling must be able to accurately predict the snowline elevation.

### 4.2 Sensitivity analysis

The scores of the contingency table analysis reveal that the thermomechanical model, which utilizes
the modeled initial and boundary conditions, can outperform a model based on calibrated guideline
friction parameters. The primary result of the preceding section is that guideline-based avalanche
dynamics models with extreme friction parameters will have difficulty reconstructing individual case
studies and that they are not easily linked to snowcover conditions. The next step is to check how
sensitive the thermomechanical model is to changes in the simulated initial and boundary conditions.





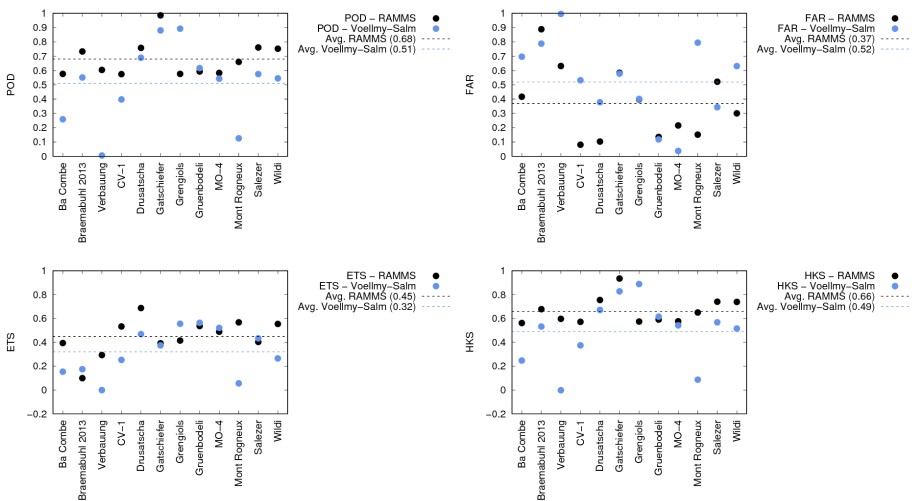

Fig. 4: Comparison of the statistical results from the thermomechanical model RAMMS (black) and the guideline-VS model (blue), for POD (a), FAR (b), ETS (c) and HKS (d).


#### 4.2.1 Role of initial conditions

To demonstrate the role of initial conditions, we simulated the twelve case studies using the initial conditions of all the other case studies, creating a total of 144 permutations. The initial conditions consist of fracture depth, snow density, temperature and LWC. For example, we simulated the Ba

Combe case study with the initial conditions from the other eleven case studies. The simulation result of every of the permutations for each avalanche path are shown in Supplement B in the online supplement.

Fig. 5 depicts the results of the 144 simulations. In these plots, the red dots indicate the sim-

ulations performed with the **SNOWPACK** modeled initial conditions belonging to the specific avalanche path; the small black dots represent the remaining combinations of eleven simulations. The large open circle represents the average of the eleven permutations.

The first result of this sensitivity analysis is that the score difference varies more than 0.2 statistical points for every avalanche path and indicator (POD, FAR, ETS and HKS scores). This result

indicates a large variability of the model with different initial conditions. The POD scores using the "'right'" initial conditions are generally higher than using those from the other case studies. Furthermore, the false alarm (FAR) rate is lower. The average of the four statistical indicators calculated with the real initial and boundary conditions (red line in Fig. 5) outperformed the calculations with




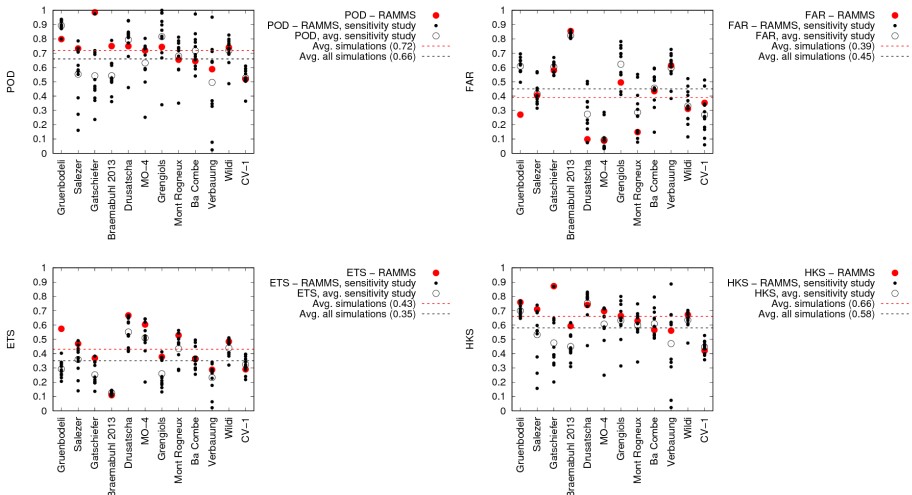

Fig. 5: Sensitivity study simulating every avalanche path with the twelve different initial and boundary conditions using the thermomechanical model RAMMS. The red dot denotes the simulation performed with the initial and boundary conditions calculated for the corresponding avalanche path. The open black circle denotes the average of the eleven permutations (filled black dots). In this plot for every avalanche path fracture and erosion depth, temperature, density and LWC at the release and along the avalanche path (erosion) are varied.

the interchanged initial and boundary conditions for every case study. However, for particular cases,

simulations with initial conditions from another avalanche path outperformed the one calculated with the real initial conditions. However, the simulation with the original initial condition is among the simulations with the highest ETS or HKS scores. Also the average scores of all twelve cases is better for the real initial conditions. A last important observation is that the spread of scores provided by the permutations of the initial conditions exceeds the spread of scores for all twelve simulations with

the real initial conditions.

Again, for the longer avalanche paths with a smooth transition to the runout zone (Gatschiefer, Drusatcha, Grengiols, Verbier Mont Rogneux and Braemabuhl), the scores varied up to 0.5 points in comparison to avalanche paths where the transition is marked by an abrupt change in slope angle (MO-4 and CV-1 and Gruenbodeli). Thus, long avalanche tracks with a smooth transition to the

runout zone, are more sensitive to changes in initial conditions and benefit most from a correct initialization using **SNOWPACK** simulations.





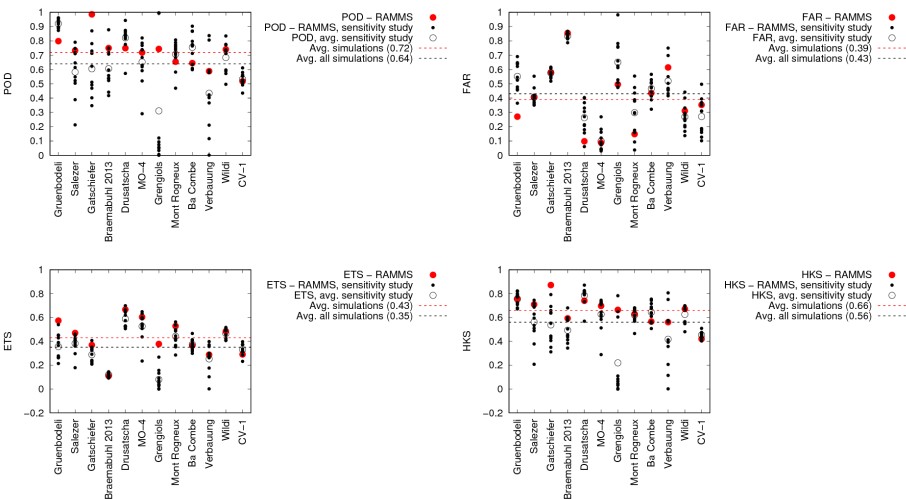

Fig. 6: Sensitivity of the thermomechanical model RAMMS to permutations of avalanche mass (fracture depth and density). For every avalanche path twelve different fracture depths, released densities, erosion depths and eroded densities are permuted, keeping the LWC and snow temperature constant. Markers and colors as in Fig. 5.

### 4.2.2 Role of snowcover mass and density

The initial conditions include both mass/density and temperature/water content. To quantify the
relative importance of initial mass versus initial snowpack properties, we performed another set of 144 simulations where only the mass (both the fracture mass and entrainment depths) varied. The results of the contingency table analysis are depicted in Fig. 6. The results are similar to the first sensitivity analysis where the entire set of initial and boundary conditions were varied. This suggests that the selection of the initial and boundary conditions for mass is more important than the
ones for temperature/LWC. For wet snow avalanches, this implies that the layers where meltwater accumulates in the release zone must be identified accurately as this defines the height of the fracture slab and therefore the release mass. A small variation in the fracture depth would lead to a large variability in the predicted avalanche runout. This is a problematic result because it indicates the critical role of fracture depth as an input parameter in avalanche simulations.

### 455 4.2.3 Role of snowcover temperature and water content

Fig. 7 displays the results of the other set of 144 thermomechanical model simulations where the temperature and LWC in the release and entrainment zones were permuted. The mass (release and eroded) was defined by the snowcover simulations driven by the meteorological data for each case




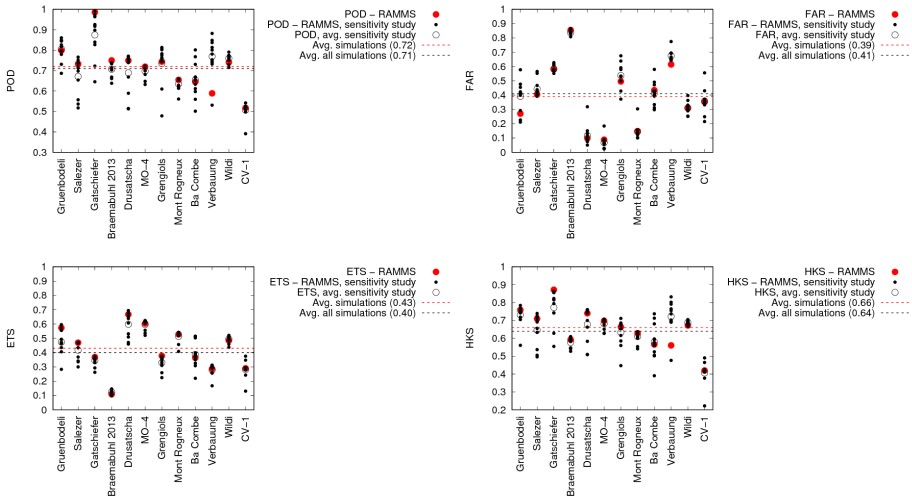

Fig. 7: Sensitivity of the thermomechanical model to different snow temperature and LWC. For every avalanche path twelve different snow temperature and LWC in the release and erosion zones are varied, keeping the release and eroded depth and density constant. Markers and colors as in Fig. 5.

study. We find the results are less sensitive to changes in temperature and LWC than to mass. This
is due to the fact that only wet snow avalanches were considered and the temperature range did not vary outside the wet snow regime. Variations are primarily due to variations in LWC. This too, is a reasonable result because moisture contents in the twelve case studies varied only between 0% and 5%, see Table 3. Although the variations are less pronounced than those caused by mass changes, Fig. 7 illustrates that correctly specifying initial snow temperature and LWC also contributes posi-
tively to the model performance. The variation was strongest on long avalanche tracks with a smooth transition to runout zone, once again indicating that this path geometry is especially sensitive to any changes in the initial conditions.

### 4.3 Sensitivity to calculation grid size

Contingency tables scores for the thermomechanical model can also depend on the selected grid
resolution. This would imply that the constant set of friction parameters of the wet snow model is bounded to a particular cell size. We subsequently repeated the simulations using three different grid sizes: 3x3 m, 5x5 m and 10x10 m. The influence on the contingency scores is depicted in Figs. 8 and 9 for 10 m and 5 m respectively.

A similar analysis was performed by (Bühler et al., 2011); however without a statistical score





and only on a limited number of case studies. The qualitative results of that study indicate that a courser resolution smooths out the terrain, causing the wet model simulations to overflow the observed deposit areas. Due to overflowing, the POD score increases by almost 0.1 statistical points in average in comparison with the 3 m resolution simulations. The coarser simulations are highly

penalized in the FAR false alarm rate indicator, showing a drop of 0.2 statistical points on average in comparison with the finer resolution. The statistical scores (ETS and HKS) were positively influenced by the increase in hit rate, but this was compensated by the even larger increase in false alarms. The ETS score is severely penalized, dropping the statistical score by 0.15 points for the coarser simulations (10 m) in comparison to finer simulations (3 m). Even though the HKS score is

more weighted to the number of hits, it likewise decreased, but by a smaller amount. The increase in false alarms was so large that it mostly compensated the improvement obtained by an increase in the number of hits.

The same analysis was repeated using 5 m resolution. In this case, the results do not differ greatly

from the results obtained with a 3 m resolution. The 5 m resolution overall statistics (see Fig. 9) are close to or even equal (in the case of the HKS score, see Fig. 5), to the results obtained by the 3 m resolution simulations. Nevertheless, the 5 m meter resolution simulations obtained higher POD score than the 3 m resolution but also a higher FAR. This pattern was already observed in the comparison between 3 m and 10 m; however, in this case the difference is much lower. In the

other two statistical indicators ETS and HKS even more similar results are obtained. The ETS score (see Fig. 9) is slightly lower for the 5 m resolution than for the 3 m. However both obtained the same score in the HKS indicator. The results obtained in the ETS and HKS indicators show the same tendency observed in the comparison between 3 m and 10 m. Coarser resolutions lead to overflowing and obtaining more hits but also more false alarms, which penalize the overall score.

Nevertheless, in the case of 3 m and 5 m, it is necessary to compare avalanche path by avalanche path and to check which resolution better suits a particular avalanche path. Narrow steep gullies with pronounced topographic features (Ba Combe, MO-4 and CV-1) require higher resolution than open slopes (Drusatscha, Mont Rogneux, Wildi and Gatschiefer).

A secondary result in this analysis, is that independent of the grid resolution, there was a large variability of the model results by varying the initial and boundary conditions. The variability found for 3 m, 5 m and 10 m cell size was similar for all case studies and for all statistical indicators.

### 4.4 Runout analysis study

A commonly used measure for avalanche size is the runout distance. Fig. 10 shows the difference

in simulated and measured runout distance for each studied avalanche for different grid cell sizes using the thermomechanical model RAMMS as well as the guidelines-VS model. The





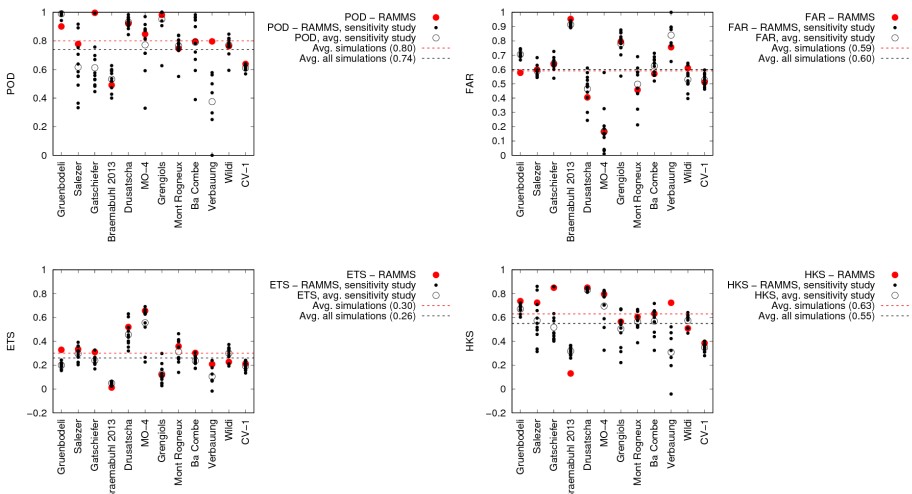

Fig. 8: Sensitivity study simulating every avalanche path with the twelve different initial and boundary conditions, but with a simulation resolution (grid size) of 10 m for the 144 simulations (compare to Fig. 5 for 3 m resolution. Markers and colors as in Fig. 5.

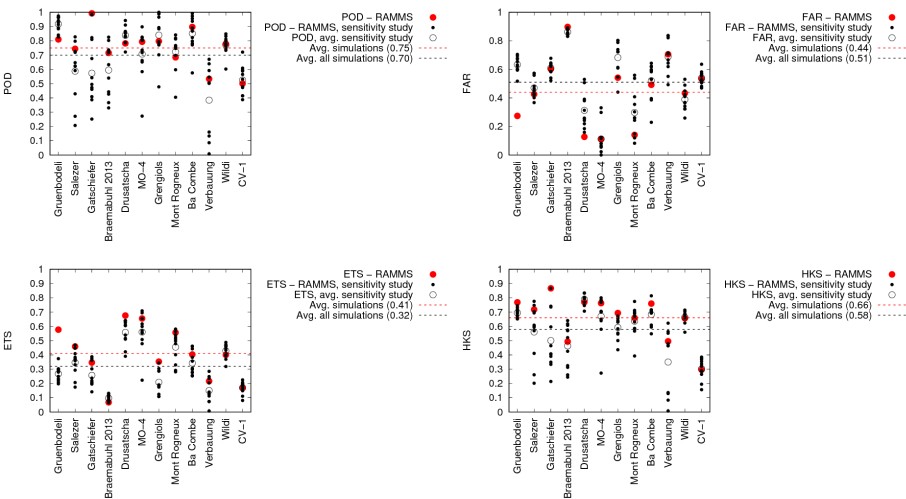

Fig. 9: Sensitivity study simulating every avalanche path with the twelve different initial and boundary conditions, but with a simulation resolution (grid size) of 5 m for the 144 simulations (compare to Fig. 5 for 3 m resolution. Markers and colors as in Fig. 5.



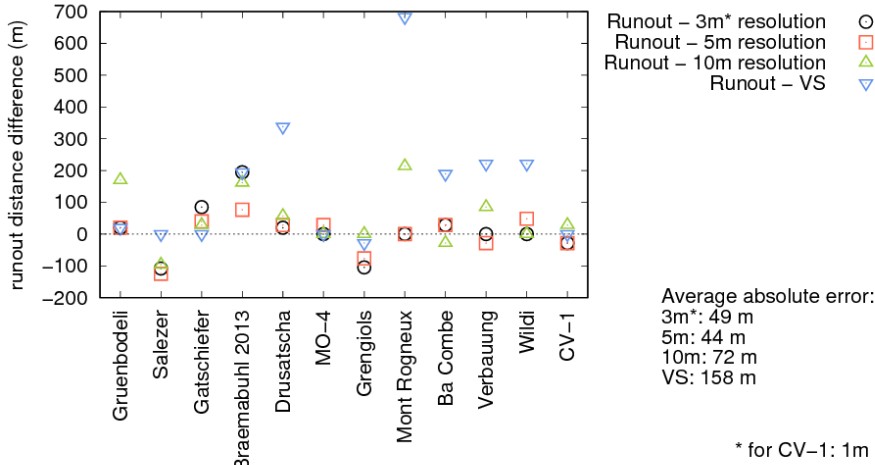

Fig. 10: Runout error plot comparing thermomechanical wet snow model calculations (black dots) with guideline-VS runout calculations (blue triangles), as well as runout calculations with 5 m and 10 m model resolution with the thermomechanical model (red squares and green triangles, respectively). The legend shows the absolute average simulation error for each set of simulations.

absolute error in runout distance calculated by the thermomechanical model is about three times smaller than those predicted by the guidelines-VS model. The difference between both models was larger on paths where the transition to the deposition zone was smoother (Drusatscha,

Braemabuhl, Mont Rogneux, Ba Combe, Gatschiefer). On the paths where this transition is more pronounced, the calculated runout distances are closer (e.g., Gruenbodeli, MO-4, CV-1, see Fig. 10).

The analysis was repeated using two coarser grid resolutions 10 m and 5 m cell size for the thermomechanical model (see Fig. 10). In the case of 10 m resolution, the model tends to overrun

measured runout distances. The average error between simulated and measured runout increases from around 49 m with 3 m resolution to 72 m with 10 m resolution. The difference between 3 m and 5 m resolution is much smaller and the 5 m resolution calculations slightly outperform the 3 m ones in terms of runout distance. On the other hand, the 3 m resolution simulations show on average higher ETS score and equal HKS score, compared to 5 m simulations (not shown).

We repeated the sensitivity study for runout distance with three sets of 144 simulations interchanging the initial and boundary conditions as described in the previous section (see Fig. 11). The results obtained performing the sensitivity analysis confirmed the results achieved in the previous contingency analysis. The thermomechanical model is sensitive to changes in the initial and boundary conditions. Those changes are more important on avalanche paths where the transition





to the runout is smooth. On those paths, changes in the initial and boundary conditions lead to deviations of hundreds of meters on runouts calculations, Gatschiefer, Drusatscha, Mont Rogneux, Ba Combe, Fig. 11.

As was shown in the contingency analysis, the runout calculations were more sensitive to changes
in mass than in changes in snowcover conditions (temperature and LWC). Varying the mass in the release and erosion doubles the absolute error obtained by varying only snow temperature and LWC (see Fig. 11).

## 5  Discussion

Our analysis is limited to evaluating deposition areas and runout distances for the twelve case
studies. Other important avalanche variables, such as speed, dynamic flow heights and impact pressures are not considered in the analysis, although they are crucial in many aspects of assessing avalanche risks. Thus, we are considering only one primary component of the avalanche flow problem: calculating the area covered by the avalanche deposits. We circumvent the lack of flow data by considering well-documented case avalanche case studies in a single flow regime (wet) with
return periods of approximately 10 to 30 years. An advantage of this approach is that we consider more than one track geometry, allowing us to draw conclusions about the application of snowcover models and avalanche dynamics calculations in different terrain. This is important because our analysis reveals that the interplay between track geometry and mass are the decisive components in the estimation of runout and inundated area.


The starting mass was specified by performing snowcover simulations to determine the fracture depth, density, temperature and water content of the release zone. The snowcover simulations were driven by measured meteorological data from stations near the release zone. The spatial extent of the release was known from observations and/or measurements. Having accurate information
where the avalanche released contributes much to the goodness of the statistical scores. Knowing the location of the release zone and a DEM of the avalanche track predetermines the flow path of the avalanche in the simulations, making a contingency table analysis useful. The model has one parameter $\alpha$ (Buser and Bartelt, 2009), which depends on the avalanche path and still has to be chosen by the avalanche expert. Therefore the application will demand experience in terrain and
modeling of avalanches by the avalanche expert, even though the range of $\alpha$ is well-constrained (Vera et al., 2016) .

An advantage of the contingency table analysis is that it can be used to identify tracks where there will be a large variability in runout depending on the initial conditions. Our analysis of the





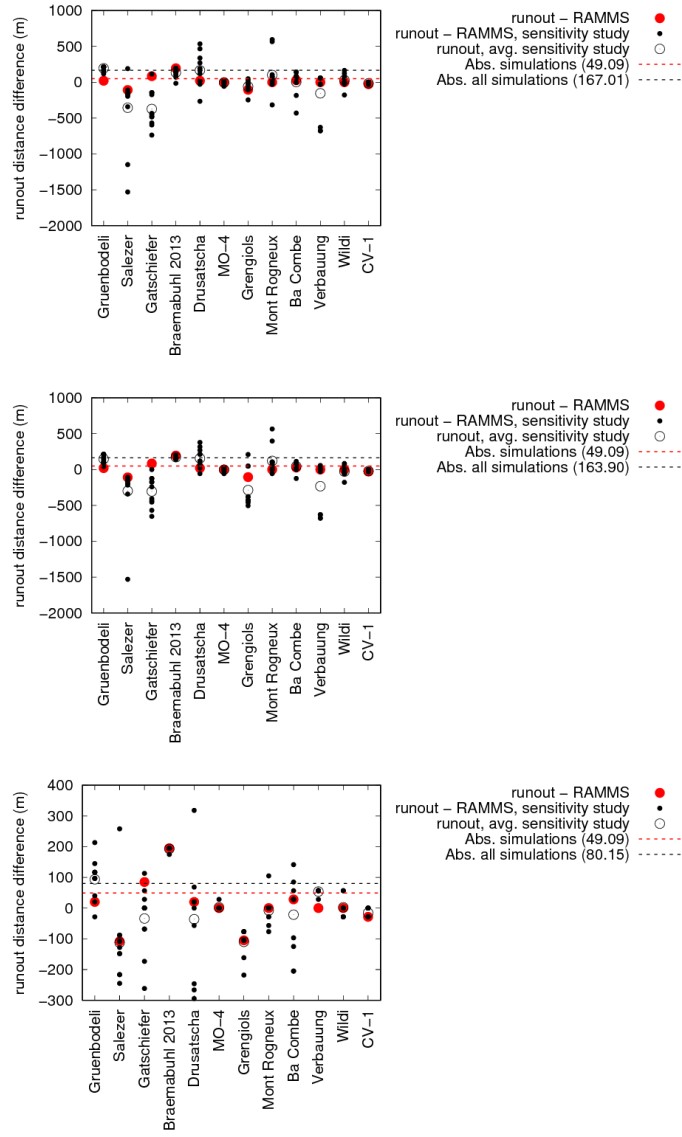

Fig. 11: Difference between simulated and measured runout distance for the wet snow model simulations with the corresponding initial conditions (red dots) and permutations (black dots). The average of the eleven permutations is depicted as a black open circle. (a) varying both snow mass (fracture depth and density) and snow properties (temperature and LWC), (b) varying snow mass only and (c) varying snow properties only. The red and black lines show the average absolute error in meters of the whole set of simulations (sensitivity and real simulations) to the runout distance measured in the field.



simulations revealed a large variability in predicted runout for tracks with flat terraces and gradual slope transitions to the runout zone. Here, we showed that the results are very sensitive to the specification of mass in the release and entrainment zones. On these tracks, an underestimation of fracture depth of only 10 cm could lead to significant runout shortening and underestimation of the affected area. However, the initial and boundary conditions estimated from snowcover

modeling have demonstrated a good accuracy in the overall results, the red dots on Figs. 5, 6 and 7 show on average better statistical scores than the black dots calculated with the variations. This result suggests statistically that initial conditions derived from snowcover modeling improve randomly chosen initial conditions derived from a set of wet snow avalanche days. Once again, although the coupling between the snowcover modeling and avalanche dynamics calculations can

be automatized, the sensitivity analysis suggests that a mistake in the mass estimation can lead to entirely wrong results. We emphasize that we come to this conclusion even though we restricted our attention to a single avalanche flow regime. Nonetheless, the coupling of snowcover models and avalanche simulations could provide avalanche services with more information to make a risk assessment. Using avalanche dynamics models in this way differs from traditional avalanche

calculations, which are based on extreme conditions, with no link to particular snowcover or meteorological conditions.

The general thermomechanical avalanche dynamics model RAMMS performs better than the guideline-VS model in all statistical scores, HKS, ETS, POD and FAR (see Fig. 4). The guideline

procedures are designed to model extreme, dry flowing avalanches, not particular avalanche events. However, the guideline model achieved in some cases high contingency table scores, despite the application on non extreme, wet snow avalanches. The guideline-VS model was forced using friction coefficients calibrated by (Salm et al., 1990). It was necessary to use the friction coefficients corresponding to smaller avalanche sizes in order to achieve a good correspondence between

measurements and simulations. For all case studies, the friction coefficients chosen correspond to size class 'Small' and a return period of 10 to 30 years. The guideline-VS model had to be manipulated by an expert user to get the best results. For example, the general model was first applied to determine the mass-balance of the event, which was then used to establish the initial conditions (i.e., released plus eroded mass) of the guideline-VS model. Another disadvantage of the

guideline model is that first a calibration of the friction parameters is required to obtain reasonable contingency table scores. Both steps are not required in the general model applications, because the friction parameters are determined as a known function of snowcover conditions. Moreover, the connection between friction and initial starting mass for the guidelines-VS model were derived from the wet snow model calculations. The guideline-VS model really cannot exploit the automated

weather measurements, and additional procedures are required to make the guidelines calculations.



Because we considered only wet snow avalanches, the range of snow temperature was rather narrow and close to zero. The water content varied between 1% and 5%, which is a typical range of bulk LWC for slopes (Heilig et al., 2015). The vertical liquid water distribution typically exhibited
a thin layer with high LWC located near layer boundaries (capillary barriers), which supports the assumption in the avalanche model that the liquid water is concentrated at the sliding surface. The results of the snowcover simulations were visually inspected to determine the avalanche fracture depth (following Wever et al. (2016)). This depth could be verified by the observations of the actual release zone. The bulk LWC of the slab above the maximum LWC was used to initialize the
simulations. In general, the statistical scores of the contingency table analysis did not change much as a function of the water content. However, changing water content in some cases led to a large difference in simulated inundation area and runout distance. These cases are associated with terrain characteristics and its influence on the rate of meltwater production as well as the LWC of the eroded snow. For example, the Grengiols and Mont Rogneux avalanche case studies stopped on a flat zone
when the initial liquid water was reduced below the simulated **SNOWPACK** value. This indicates that underestimated LWC can lead to spurious runout shortening. In general, however, variations of mass (i.e., fracture and erosion depths together with snow density) produced larger variations in the final simulation results (see Fig. 5, 6 and 7). The mass variations in the sensitivity analysis were broad, see Table 1. Therefore, using this set of case studies with only wet snow avalanche cases, the
model is more sensitive to changes in avalanche mass than in snowcover conditions (LWC and snow temperature).

The statistical scores of the contingency table analysis are dependent on the grid resolution of the avalanche dynamics calculations. The 10 m resolution appears to be far too coarse for the avalanche sizes of the case study examples. The contingency scores of the 3 m and 5 m resolutions are similar.
However, the 3 m runout calculations show a trend to slightly shorter runout distances. The statistical scores of the 3 m resolution are overall better than the 5 m resolution because the 3 m scores were not penalized by excess runout and therefore obtained fewer false alarms. The 5 m resolution clearly achieved the best results for open slopes with gradual transition zones. A 3 m resolution might still be necessary when the track contains narrow gullies, bare ground or shallow snowcovers where terrain
features, including the presence of blocky scree, can play an important role. Deposition patterns of the smaller events could clearly be better represented by the finer 3 m resolution.

## 6   Conclusions

We used the physics based snowcover model **SNOWPACK** to set the initial conditions for avalanche dynamics calculations. We restricted our attention to avalanches in one flow regime (wet) where the
depth and spatial extent of the avalanche release area was known. We used a contingency table analysis to statistically evaluate how well avalanche dynamics models can predict deposition area



and runout distances. Although we can demonstrate that physics based models improve the statistical scores, we note that on certain track geometries the results of the avalanche dynamics calculations are extremely sensitive to the specification of the correct starting conditions, particularly fracture
and entrainment depths. These tracks contain flat track segments below the release zone and gradual transition zones leading towards the avalanche runout zone. In these cases, underestimating fracture heights and entrainment depths can lead to significant underprediction of avalanche runout distances. The problem appears not to be with the quality of the avalanche dynamics simulations, but illustrates that for these cases it is crucial that numerical snowcover models accurately predict the state of the
snowpack from data measured from automatic weather stations.

The model chain could be applied in regions where considerable experience and knowledge of local snowcover variability and avalanche history exist. As these conditions change from year to year, a complete cadaster of documented events is still invaluable. There are cases where these conditions are fulfilled, see Vera et al. (2016). In these situations the model chain can support
decisions on a deterministic basis and provide decision makers with a valuable source of information about current avalanche risks.

*Acknowledgements.* Financial support for this project was provided by Codelco Mining, Andina Division (Chile). We thank all Codelco avalanche alert center members L. Gallardo, M. Didier and P. Cerda, together with the Mountain Safety crew not only for their support, but also for their confidence, patience and enormous
help during the last four winters in the Andina mine.





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
