# Peer review of "Modeling the influence of snowcover temperature and water content on wet snow avalanche runout"

_Natural Hazards and Earth System Sciences, 2017_

## Referee Comment (RC1)

**Review of 'Modeling the influence of snowcover temperature and water content on wet snow avalanche runout' by C. VERA VALERO et al.'**

In this paper, the authors present a model chain for the back calculation of twelve well documented (mostly wet) snow avalanches. The snowcover simulation model "SNOWPACK" is used to derive snow cover properties as input data (release and model parameters) for avalanche simulations. Avalanche simulations are performed with the toolbox "RAMMS", employing a classical flow model with Voellmy friction relation and an extended thermomechanical flow model. Different statistical scores are introduced to evaluate the simulation performance regarding the comparison of simulated flow depths and documented deposition patterns. With these statistical scores and runout estimates the simulation sensitivity is investigated with respect to different kinds of input sources (simulation input, model parameters, grid resolution). Topic and content of the paper fit well to the audience of NHESS. However the reader may be confused because important links and a central theme seems to be missing. A possible solution to finalize this paper could be to either concentrate on one of the three main subjects or to somehow relate them in a consistent way.

The presented model chain consists of the two components: a *snow cover model*, which runs on measured meteo data and *avalanche simulations*, which use the snow cover properties provided by the snow cover model. Statistical scores and runout comparison appear as very useful tool to objectively evaluate the avalanche simulation, i.e. the last part of the model chain - variations of snow cover model performance and variability are not presented. The analysis can be divided in three main (somehow mixed but independent!) contributions: (i) model chain performance check and cross comparison to the classical approach, (ii) sensitivity analysis of the thermomechanical avalanche simulations with respect to avalanche path location (model input parameters), (iii) avalanche simulation sensitivity analysis with respect to computational/terrain resolution. Although the presented approaches appear to be highly interesting and promising some parts are incomplete or at least not well structured/distinguished. Throughout the paper there is a need to clarify what (and why) the authors exactly do:

**general questions:**

- What is the main goal of the analysis? A new simulation evaluation approach? Introducing or testing a new flow model? Sensitivity study with respect to grid resolution?

- What exactly is deposition in terms of simulation results (deposition is not directly modeled in RAMMS, hence 20cm flow depth are compared to observed deposition, but when does an avalanche simulation stop? why is this an appropriate choice?). Why is the runout analysis separated from the statistical scores and not equally treated?

- What is the advantage of four different statistical scores, when they are based on two independent measures that could deliver the same general message (variation of simulation results)?

- How are simulation input, model parameters, boundary and initial conditions distinguished (e.g. density is a snow cover property in terms of snowpack simulation, describing the release mass and also a flow model parameter in terms of avalanche simulations?)?

- Is section 2 needed or would it be more beneficial to discuss the evaluation approach in more detail and simply refer to Valero et al. (2016)?

- How does the snow cover simulation perform in comparision to field data (e.g. field observations on fracture depths, densities, . . . )?

Overall the manuscript is well written and the derived figures 4-9 appear useful to interpret the statistical outcome of the sensitivity analysis. However, for better comparability, the figure axis should have the same limits (e.g. HKS of figures 6-7). Same holds for the figures in supplemental material (e.g.

supp. A, figure S8 a-b). Generally it should be stated what exactly is shown in the supplement figures (A+B) (deposit depth is not a direct simulation result - is it flow depth at a certain (which?) time step? What is depicted by the red outlines (which are very hard to distinguish) in supplement B (20 cm flow depth outlines?)?).

**specific questions:**
  **In section 3.2.** ((i) model chain performance check and cross comparison to the classical approach) the authors outline their performance evaluation strategy (guideline parameters with classical flow model vs. modeled snow pack properties + ad hoc parameter assumptions for the new thermomechanical flow model). It appears that some crucial questions remain unclear:

- Can the simulation approaches really be compared like this? Is this a comparison of simulation strategies/procedures or of flow models?

- Why does it make sense to use a mix of modeled snow cover parameters (depth) and guideline parameters?

- The growth indices should depend on the choice of flow and the entrainment model/parameters, so they are a result of the model chain?

- With the thermomechanically modelled growth indices, the initial mass of the classical simulations are set. But since classical VS parameters where *calibrated* implicitly including entrainment (field observations that include entrainment) this should not be necessary?

- Why is it appropriate to choose model parameters for "small, frequent" avalanches for all events, when release volume of e.g "Gatschiefer" is up to 330.000 m$^3$?

- How can the ad hoc model parameter choices for the thermomechanical model be justified (that vary for different avalanche paths, e.g. Entrainment coefficient (0.6-0.8) and $\alpha$ parameter)? Or does the choice not matter because the result influence is negligible?

  **In section 3.5** ((ii) sensitivity analysis of the extended avalanche simulations with respect to avalanche path location) three different approaches to study the sensitivity of the thermomechanical model are performed (interchanging all or combinations of the model parameters that are related to the snow cover model - fracture and erosion depths, density, snow temperature and LWC). The sensitivity analysis is evaluated on a qualitative level, e.g. no single parameters ranges are investigated (varied with respect to their absolute values) with a quantification of the output variability (which would actually be the advantage of the introduced statistical measures). Open questions are:

- The snow cover model parameters are permuted by event location. With this no quantitative evaluation is possible with respect to the absolute variation of avalanche simulation input, which are (as depicted in table 2) ≈26% for release depth, ≈16% for densities and ≈46% for LWC and ≈151% for temperatures (compared to the respective mean value). Considering these differences (in magnitudes) a direct, systematic comparison and sensitivity analysis is hardly possible - how can we finally conclude which parameters are more important if the are not *equally* treated?

  **In section 4.3** ((iii) avalanche simulation sensitivity analysis with respect to computational/terrain resolution) the sensitivity with respect to grid size is evaluated. Main questions are:

- As i understand it - this analysis treats the computational grid resolution. How is the DEM resolution treated (resampled to the computational resolution)?

- The main result is that the presented method (statistical scores) can show that parameter values are bound to certain spatial resolutions. Since this has been observed before (e.g. by Bühler et al., 2011, as stated by the authors) this section could maybe be moved to the appendix to smooth the entire manuscript.

**Please find some more detailed line-by-line comments/questions below:**

- *title* The title of the paper **Modeling the influence of snowcover temperature and water content on wet snow avalanche runout** could focus more on the main contributions (simulation evaluation/sensitivity) and results of the paper (as stated in the abstract *Reliable estimates of avalanche mass (depth and density) in the release and erosion zones is identified to be more important than an exact specification of temperature and water content.* - which slightly contradicts the title).

- *abstract* Do height and depth have different meanings? Is it consistent throughout the paper?

- *3, 66, ... deposits area ...* deposition area prediction

- *3, 67, Instead of parameter optimization, ...* This is a crucial point. If you pursue a flow model comparison, both models should be equally treated, i.e. performing a full optimization and comparing the result performance, not to compare apples and oranges (c.f. Rauter et al., 2016, where a extended flow model is also compared to a Voellmy friction relation with different measures). If you pursue a comparison of simulation approaches/strategies, guidelines should not be mixed with model chain results.

- *3, 74, 3. ..., Fig. 1* To me it appears that the "model chain" is the combination of snowpack and avalanche simulations. The statistical scores/analysis is valid tool to evaluate the results (jointly with the runout estimates) but not a part of the chain. Similar evaluations have been performed for operational avalanche simulations Naaim et al. (2013) (snow properties and simulated avalanche runout) or Fischer et al. (2015) and recently for other mass flows Mergili et al. (2017) (introducing statistical scores to evaluate model performance).

- *Section 2: Wet snow avalanche modeling* In this section the underlying avalanche flow model is described. Since it corresponds to Valero et al. (2016) it could be omitted or transferred to the avalanche dynamic modeling (section 3.2 or appendix) part, as it distracts from the main topic of this paper.

- *9, 219, ... apply a three-dimensional avalanche dynamics model* Maybe better: Two dimensional model operating in three dimensional terrain.

- *9, 221-223, The small elevation difference between the release zones and the weather stations ... provides the sufficient conditions to ...* What do you mean with "sufficient conditions", i.e. sufficiently small?

- *13, 294, class Small avalanches.* Same class for all release volumes from 4.000 m$^3$ up to 330.000 m$^3$ - is this in correspondence to (Salm et al., 1990)?. There are also reasons to assume that no mass/volume dependency is necessary and that parameters cannot be interchanged between locations (especially regarding non extreme events, c.f. Issler et al., 2005; Gauer et al., 2010).

- *13, 298, Section 3.3 Contingency table analysis for deposition area* How do these scores compare to similar approaches evaluating snow avalanche simulations (Fischer et al., 2015; Rauter et al., 2016) and other mass flow simulations Mergili et al. (2017)? Would it also be possible to show the result variability with only two of the scores (since they are based on two independent measures)?

- *14, 316, section 3.4 Avalanche runout* This is an interesting definition of runout in a simulation framework - what are the advantages and disadvantages of this definition (are there limitations for multipath effects?, c.f. Fischer, 2013)? Some more details on how the final time steps or simulation patterns are determined would be interesting (dependence on numerical parameters, e.g. cut off for flow depths? what are the stopping criteria/simulation times?, c.f. Teich et al. (2014)?).

- *14, 323, section 3.5 Influence of initial conditions on avalanche runout: sensitivity study* **and** *17, 403, section 4.2 Sensitivity analysis* The intention of an objective sensitivity analysis seems promising, but a systematic approach, which leads to clear and quantifiable results regarding the influence of single parameter/input variables is missing (see general questions above). The general result, that interchanging model parameters from one event to another, reduces the simulation performance is not surprising.

- *16, 372,* the guideline-VS model. The

- *25, 540, . . . such as speed, dynamic flow depths . . . .* Is it possible to give an estimate on the magnitude of their variability?

- *25, 540-542, . . . avalanche risks.* What would be the benefit of using further modeling results? Why is it not necessary to consider them ( compare, e.g. for avalanche velocities Sailer et al. (2002); Ancey and Meunier (2004); Gauer (2014) or Sovilla et al. (2007); Fischer et al. (2015) for growth indices/mass balance)?

- *26, figure 11* For better comparability the same scaling of the y-axis of the single figures (a), (b) and (c) would be desirable.

- *28, 635, . . . depth and spatial extent of the avalanche release area was known.* How does the SNOW-PACK model perform regarding the documented release depths - are there any measurements available?

**References**

Ancey, C. and Meunier, M. (2004). Estimating bulk rheological properties of flowing snow avalanches from field data. *Journal of geophysical research*, 109(F01004):doi:10.1029/2003JF000036.

Bühler, Y., Christen, M., Kowalski, J., and Bartelt, P. (2011). Sensitivity of snow avalanche simulations to digital elevation model quality and resolution. *Annals of Glaciology*, 52:72–80, doi:10.3189/172756411797252121.

Fischer, J.-T. (2013). A novel approach to evaluate and compare computational snow avalanche simulation. *Natural Hazards and Earth System Science*, 13(6):1655–1667.

Fischer, J. T., Kofler, A., Fellin, W., Granig, M., and Kleemayr, K. (2015). Multivariate parameter optimization for computational snow avalanche simulation. *Journal of Glaciology*, 61(229):875–888.

Gauer, P. (2014). Comparison of avalanche front velocity measurements and implications for avalanche models. *Cold Regions Science and Technology*, 97(0):132 – 150.

Gauer, P., Kronholm, K., Lied, K., Kristensen, K., and Bakkehøi, S. (2010). Can we learn more from the data underlying the statistical $\alpha - \beta$ model with respect to the dynamical behavior of avalanches? *Cold Regions Science and Technology*, 62:42–54.

Issler, D., Harbitz, C., Kristensen, K., Lied, K., Moe, A., Barbolini, M., De Blasio, F., Khazaradze, G., McElwaine, J., Mears, A., Naaim, M., and Sailer, R. (2005). A comparison of avalanche models with data from dry-snow avalanches at Ryggfonn, Norwaykav. *Proc. 11th Intl. Conference and Field Trip on Landslides, Norway, 110 September, 2005. Netherlands, A. A. Balkema, Taylor & Francis Group*, pages 173–179.

Mergili, M., Fischer, J.-T., Krenn, J., and Pudasaini, S. P. (2017). r. avaflow v1, an advanced open-source computational framework for the propagation and interaction of two-phase mass flows. *Geoscientific Model Development*, 10(2):553–569.

Naaim, M., Durand, Y., Eckert, N., and Chambon, G. (2013). Dense avalanche friction coefficients: influence of physical properties of snow. *Journal of Glaciology*, 59(216):771–782.

Rauter, M., Fischer, J.-T., Fellin, W., and Kofler, A. (2016). Snow avalanche friction relation based on extended kinetic theory. *Natural Hazards and Earth System Sciences*, 16(11):2325–2345.

Sailer, R., Rammer, L., and Sampl, P. (2002). Recalculation of an artificially released avalanche with SAMOS and validation with measurements from a pulsed Doppler radar. *Natural Hazards and Earth System Sciences*, 2:211–216.

Salm, B., Burkhard, A., and Gubler, H. U. (1990). Berechnung von Fliesslawinen: Eine Anleitung fuer Praktiker; mit Beispielen. *Mitteilungen des Eidgenoessischen Instituts fuer Schnee- und Lawinenforschung*, 47:1–37.

Sovilla, B., Margreth, S., and Bartelt, P. (2007). On snow entrainment in avalanche dynamics calculations. *Cold Regions Science and Technology*, 47(1-2):69–79.

Teich, M., Fischer, J.-T., Feistl, T., Bebi, P., Christen, M., and Grêt-Regamey, A. (2014). Computational snow avalanche simulation in forested terrain. *Natural Hazards and Earth System Science*, 14(8):2233–2248.

Valero, C. V., Wever, N., Bühler, Y., Stoffel, L., Margreth, S., and Bartelt, P. (2016). Modelling wet snow avalanche runout to assess road safety at a high-altitude mine in the central andes. *Natural Hazards and Earth System Sciences*, 16(11):2303.

---

## Referee Comment (RC2) · G. Chambon (Referee) · 6 Sep 2017

I commend the authors for the impressive amount of work summarized in this paper: the compilation of data, systematic SNOWPACK and RAMMS simulations, and extensive sensitivity study provide an unprecedented set of results concerning the modelling of wet snow avalanches and the influence of various parameters such as initial mass and snow temperature / LWC on avalanche deposits and runouts. Despite the complex chain of models that is used, the authors made the effort to try to isolate the most influential physical processes, which I find particularly interesting. I am henceforth fully favorable to the publication of this paper in NHESS. I think however that several aspects

of the paper could be improved to provide a better account of this nice study. First, the paper is a bit lengthy and redundant at places, and the structure of certain sections could be improved. Most importantly, I feel that the choice made by the authors to base most of the discussions on the statistical scores coming from the contingency table analysis, sometimes tend to "soften" the results and "dilute" the differences among the models. Putting more emphasis on more "physical" outputs, such as the raw results shown in supplementary material and the runout distances, would help counterbalance this trend. Finally, I consider that the discussion of the sensitivity analysis needs to be complemented with more quantitative comparisons and discussions. The specific comments below provide more detailed suggestions on these issues.

Specific comments

1/The introduction would benefit from being more to the point at certain places. The second paragraph, in particular, appears a bit off-topic and overly speculative. If the goal is to explain that wet snow avalanches are characterized by relatively large values of apparent viscosity and cohesion, there is probably no need to discuss the so-called "compactive strength" of snow and its hypothetical relation with viscosity. On the other hand, in the third paragraph, a more in-depth discussion of the advantages and drawbacks of the different approaches used in past studies to model wet snow avalanches would be in order.

2/Section 2, presentation of the model: A clearer structure (e.g., avoiding redundancies and introducing subsections / subtitles to better distinguish between the different elements of the model) would improve the readability of the section. Moreover, certain mathematical notations could probably be simplified, and some physical relations better explained. Some suggestions below:

-Why using the subscript $\Phi$ everywhere? Is it really useful?

-The variable $N_K$ present in Eq. (1) would need to be defined earlier after this equation.

-What is the parameter $\gamma$ in Eq. (3)?

-What are the quantities $h_\Phi^s$ and $\rho_\Phi^s$ in line 117?

-Indicate the physical meaning of $S_\Phi$ (shear stress).

-The sentence starting with "The basal boundary converts..." on line 129 is not very clear. This point would maybe be better explained in conjunction with Eq. (7)?

-What is the relation between the quantities $\dot{P}_\Phi$ and $R_\Phi$? Why not denoting the former simply as $\dot{R}_\Phi$

-Idem: what is the relation between $\dot{P}_\Phi^V$ and $R_\Phi^V$?

-What is the coefficient $c$ in Eq. (11)?

-The sentence starting with "Equation (14) takes into account..." in line 174 is not very clear.

-The specific form chosen for the cohesion, i.e. the factor $(1-\mu)$ and the exponential term, should be commented.

3/Section 3.1. It is not fully clear whether SNOWPACK simulations were performed only for the release zones, or also for the deposition zones (in cases where data are available for these zones).

4/Section 3.1, Table 3. How is the erodibility coefficient obtained? This parameter is not discussed in the text, although its influence on the results is probably far from negligible.

5/Section 3.2. The value chosen for the parameter $\zeta$ involved in Eq. (7) should also be discussed .

6/Section 3.2. Besides data on avalanche release area, the authors probably also have data on fracture depths for at least some of the avalanches. How do these data compare to the fracture depths predicted by SNOWPACK? Where they used in any

way to optimize the results of SNOWPACK?

7/Section 3.2. Regarding the Voellmy-Salm model, and if I understood well, the authors chose to use the same friction parameters for all studied avalanches. Would not it make more sense to optimize these parameters for each avalanche? I do not see any reason why all these avalanches should be characterized by identical friction parameters. In addition, giving the value of these parameters would also be useful, for the sake of comparison with the parameters used in the RAMMS model.

8/Section 3.5 is not very clear and some redundancies could be avoided. In the first paragraph, in particular, it is difficult to understand what the 432 simulations represent, whereas this issue is better explained afterwards.

9/Section 4.1. While the different statistical scores used by the authors effectively show that the thermomechanical model performs better than the Voellmy-Salm model, this issue is even more evident from observation of the model outputs provided as supplementary material. Hence I would encourage the authors to add, at least, a short description of these raw outputs in the main text prior to discussing the statistical scores. Adding a figure showing one or two illustrative examples of raw results in the main text could also be option. Similarly, moving the runout comparisons (currently presented in 4.4) before the statistical score comparisons could also help to better illustrate the differences among the models.

10/Section 4.1. The sentence starting by "The fact that the difference in ETS score..." in line 379 seems in contradiction with what is said just before (lower difference in ETS than in HKS between the two models).

11/Section 4.2, line 407. Why do the authors refer to the friction parameters used in the VS model as "extreme" here?

12/Section 4.2-4.3-4.4. I encourage the authors to provide more quantitative evidences of the conclusions drawn from their sensitivity study. In the current manuscript, it is

sometimes difficult to relate the assertions made in the text to the presented data. One probable reason is that the authors rely throughout on the same type of figures, whereas alternative representations, such as boxplots or distributions / percentiles, would probably allow for easier quantitative comparisons between, e.g., the different initial conditions (mass versus temperature/LWC) or the different grid resolutions. I indicate below a few examples of overly qualitative statements that would need to be supported by more quantitative evidences:

-line 426: "generally higher"

-line 431: "the simulation with the original initial condition is among. . ."

-line 440: "are more sensitive to"

-line 452-423: "A small variation (. . .) would lead to a large variability" (While Fig. 6 shows that simulations with other initial conditions are sometimes as good as simulations with the correct initial conditions.)

-line 459: "less sensitive"

-line 465: "The variation was strongest"

13/Section 4.2.2: Could the authors also discuss the relative influence on the results of mass in the release area versus mass in the entrainment zone?

14/Section 4.3: The description of the effect of grid size on the statistical scores could probably be shortened, and redundancies avoided. I suggest however to extend the – currently very short – last paragraph describing the interplay between initial conditions and resolution. To me, this latter issue constitutes the real novelty of the sensitivity study conducted by the authors with respect to grid resolution.

15/Section 5: The sentence starting with "Moreover, the connection between friction and initial starting mass" in line 597 is not very clear.

Minor issues

-Table 1. The caption mentions virtual slope, but this information does not seem to appear in the table?

-Line 249-252: The sentence starting with "In case of avalanches with new snow. . ." is not fully clear: does it apply only to the cases where meteorological data in the deposition zone are not available, or to all cases?

-Line 308: The reference to Table 2 seems wrong here.

-Fig. 3, caption: word missing after "the longest calculated".

-line 477: typo: "courser"

-Fig. 10: Why the asterisk with the specific value corresponding to the CV-1 case?

-line 524: why the "(not shown)", instead of a reference to section 4.3 where variations of ETS and HKS with resolution are extensively discussed?

-Fig. 11: (a), (b), (c) need to be added to the plots.

-line 609: word missing after "the maximum LWC"?

---

## Author Response (AR1)

**Response to review by J.-T. Fischer**

*In this paper, the authors present a model chain for the back calculation of twelve well documented (mostly wet) snow avalanches. The snowcover simulation model SNOWPACK is used to derive snow cover properties as input data (release and model parameters) for avalanche simulations. Avalanche simulations are performed with the toolbox RAMMS, employing a classical flow model with Voellmy friction relation and an extended thermomechanical flow model. Different statistical scores are introduced to evaluate the simulation performance regarding the comparison of simulated flow depths and documented deposition patterns. With these statistical scores and runout estimates the simulation sensitivity is investigated with respect to different kinds of input sources (simulation input, model parameters, grid resolution). Topic and content of the paper fit well to the audience of NHESS. However the reader may be confused because important links and a central theme seems to be missing. A possible solution to finalize this paper could be to either concentrate on one of the three main subjects or to somehow relate them in a consistent way. The presented model chain consists of the two components: a snow cover model, which runs on measured meteo data and avalanche simulations, which use the snow cover properties provided by the snow cover model. Statistical scores and runout comparison appear as very useful tool to objectively evaluate the avalanche simulation, i.e. the last part of the model chain - variations of snow cover model performance and variability are not presented. The analysis can be divided in three main (somehow mixed but independent!) contributions: (i) model chain performance check and cross comparison to the classical approach, (ii) sensitivity analysis of the thermomechanical avalanche simulations with respect to avalanche path location (model input parameters), (iii) avalanche simulation sensitivity analysis with respect to computational/terrain resolution. Although the presented approaches appear to be highly interesting and promising some parts are incomplete or at least not well structured/distinguished. Throughout the paper there is a need to clarify what (and why) the authors exactly do: general questions:*

**ANSWER:** We thank Dr. Fischer for his review and very constructive and helpful comments. We changed the abstract and introduction to make our motivation more clear.

**ANSWER:** The first version of the paper did not contain a description of the model. We abandoned this version because we could not rationally describe the simulation results and statistical analysis without refering to model input and output. Section 2 serves to define what the model input is, and what the model produces. It is central in understanding the model chain. We stress the goal of the paper is not to make a model comparison, or to present a method of statistical analysis. The goal of the paper is to identify what boundary conditions MUST be ACCURATELY specified in order to produce reliable simulation results. We found that SNOWPACK can be used, however, there are difficulties. Our approach is to keep the avalanche dynamics parameters (more-or-less) constant, but specify the initial and boundary conditions based on SNOWPACK simulations. The model description serves to help the reader distinguish between material parameters (for wet snow) and initial and boundary conditions. Without the model description we found that it was impossible to clearly separate the two. For this reason we need a description of the model. We perfectly understand the comments of the reviewer. It is not our purpose to write a long paper; but without the model description it is impossible for the reader to judge the results of the simulations, and therefore the model chain. Our goal is to present the entire model chain, from snowpack simulations (which we don't describe in detail), through avalanche dynamics simulations and statistical analysis. In order to appease the reviewer, we have restructured the model description, allowing the interested reader to read only those parts of interest.

**CHANGED:** The second reviewer also had difficulties with our motivation. We revised the introduction to clearly state the main goal of the paper. For the sake of clarity we also restructured the model description. See the track changes file.

- *What is the main goal of the analysis? A new simulation evaluation approach? Introducing or testing a new flow model? Sensitivity study with respect to grid resolution?*

**ANSWER:** The goal of the paper is NOT a new simulation evaluation approach. The goal of the paper is NOT to test a flow model. Our goal is to pinpoint the primary difficulty of modelling wet snow avalanches. Our goal is to show that accurate boundary conditions are

necessary for thermomechanical avalanche dynamics models. However, We come to the somewhat surprising sub-conclusion: "Reliable estimates of avalanche mass (height and density) in the release and erosion zones is identified to be more important than an exact specification of temperature and water content." Moreover: we come to the conclusion that snowcover models must be able to identify where meltwater accumulates (this defines the amount of release mass.) This is the result that we want to bring forward. We clearly state this goal and result in the abstract. The evaluation approach, which we do not consider new, is used to support this claim. We repeat: we DO NOT want to develop a general method to evaluate model results.

CHANGED: We revised the introduction to clearly state the main goal of the paper. For the sake of clarity we also restructured the model description.See the track changes file

- *What exactly is deposition in terms of simulation results (deposition is not directly modeled in RAMMS, hence 20cm flow depth are compared to observed deposition, but when does an avalanche simulation stop? why is this an appropriate choice?). Why is the runout analysis separated from the statistical scores and not equally treated?*

ANSWER: We agree with Dr. Fischer that it is not appropriate to compare 20cm flow depth with measured depositions of 1m or 2m. We added the sentence "Variations in modelled and observed deposition heights are not captured with this procedure" in section 3.3 to clarify our approach and address the concerns of the reviewer. Our philosophy is to adopt a practical approach: as a first step we comparted the measured and simulated inundation areas, independent of the deposition heights. This is how the simulation models would be applied in practice. We admit that the measured and simulated flow heights WITHIN the deposition area might differ (the reviewer is CORRECT), but suggest the first necessary step is to compare the measured and calculated inundation areas. The models are simply NOT that accurate (yet) to make deposition height comparisons, which are often a function of very, very local conditions. This is why we restrict the paper to the inundated areas. Regarding the comment on runout distance: runout distances provide an intuitive measure. It is also a variable that is used avalanche classification systems. For this reason, we are motivated to show our results for runout distance. The contigency table analysis needs four classes that can only defined in a two-dimensional terrain analysis. Runout, on the other hand, provides only a binary result, hit or miss and therefore NO statistical score. We placed the sentence "This two-dimensional procedures avoids the problem of defining a one-dimensional measure of avalanche runout" in section 3.3.

- *What is the advantage of four different statistical scores, when they are based on two independent measures that could deliver the same general message (variation of simulation results)?*

ANSWER: No, they do not deliver the same general message. We consider all FOUR statistical scores to be relevant and necessary. Again, this has to do with the use of inundation areas to describe model performance. We simply want to know when the simulated model results are correct (hits), or when they predict inundated areas where they were not (false alarms). The HKS and ETS are summarizing statistics, giving an overall statistical score, but they don't allow for much interpretation. A low HKS may result from low probability of detection or high false alarm rate. Note that it is important not to use only probability of detection, as it is easy to cheat this score: just make the avalanche as large as possible and you'll optimize the probability of detection. For this reason, most studies using contigency table analysis show multiple statistical scores, to allow for interpretation of the scores.

- *How are simulation input, model parameters, boundary and initial conditions distinguished (e.g. density is a snow cover property in terms of snowpack simulation, describing the release mass and also a flow model parameter in terms of avalanche simulations?)?*

ANSWER: We include Table **2** and **3** refer to boundary conditions (snow properties for release and erosion); Table **4** shows the model parameters, for the guidelines and the thermomechanical models. Snow properties are supplied by the SNOWPACK model. It would be

really nice to have measure snow properties everywhere, but this is simply an impossibility. Second, we apply ONLY ONE set of friction parameters (those for wet snow avalanches). We change snow properties (initial and boundary conditions) but do NOT change model parameters. This is one reason why we describe the model in section 2 to distinguish between the model parameters and the snow properties. The reader should obtain this information by reading sections 2 and 3. This is why we want to keep the model description in order to clearly identify what the difference snow properties and model paramters.

- *Is section 2 needed or would it be more beneficial to discuss the evaluation approach in more detail and simply refer to Valero et al. (2016)?*

ANSWER: We restructured the modelling section. A first draft of the paper did not include the model description. We found, however, that when discussing the results that physical knowledge of wet snow avalanche modelling is needed. For example, we cannot talk about LWC without defining how LWC is included in the avalanche dynamics model. These initial conditions (based on physical modelling) need to transferred to the avalanche dynamics model. Again, the purpose of the paper is to highlight what we regard to be an important PHYSICAL result: we need to know where meltwater accumulates with the snowcover (e.g. base or interior layer) to establish the initial conditions of the simulation. This problem, which is immense, might exclude the application of avalanche dynamics models to perform "real time hazard mapping" in future. Because our results QUESTION the application of models, we believe the model and model performance should be presented in the same paper. CHANGED: We restructured the model description.See the track changes file

- *How does the snow cover simulation perform in comparision to field data (e.g. field observations on fracture depths, densities, . . . )?*

ANSWER: For a few case studies, additional information is available for fracture depths, (laser scan and drone measurements in some of the case studies. see modified Table 1. There have been many papers written validating the SNOWPACK model. In the manuscript there are five references that validate the performance of the snow cover model. Three of them were written by one of the co-authors and are related to wet snow modelling using field data from the area where 7 of our case studies occured. We consider that the snow cover model was tested enough and we do not consider this manuscript as the place to re-evaluate the snow cover model performance. CHANGED: We revised Table 1 to include fracture depths where possible.

*Overall the manuscript is well written and the derived figures 4-9 appear useful to interpret the statistical*

*outcome of the sensitivity analysis. However, for better comparability, the figure axis should have the same limits (e.g. HKS of figures 6-7). Same holds for the figures in supplemental material (e.g. supp. A, figure S8 a-b). Generally it should be stated what exactly is shown in the supplement figures (A+B) (deposit depth is not a direct simulation result - is it flow depth at a certain (which?) time step? What is depicted by the red outlines (which are very hard to distinguish) in supplement B (20 cm flow depth outlines?)?).* ANSWER: The supplement has been modified. Thank you for the suggestions and we apologize for the omission of a complete figure description in the supplement. In Supplement B, the color bar denotes deposits height (i.e., flow height in last time step $> 20$ cm) of the simulation with the initial conditions corresponding to the event. The outlines of the simulated deposits (i.e., flow height in last time step $> 20$ cm) for each of the other 11 different initial conditions are shown in varying degrees of rosa to red color. CHANGED: We have modified the figures and the supplement according with your suggestions.

**specific questions:**

*(In section 3.2.(i) model chain performance check and cross comparison to the classical approach) the authors outline their performance evaluation strategy (guideline parameters with classical flow model vs. modeled snow pack properties + ad hoc parameter assumptions for the new thermomechanical flow model). It appears that some crucial questions remain unclear:*

- *Can the simulation approaches really be compared like this? Is this a comparison of simulation strategies/procedures or of flow models?*

**ANSWER: We emphasize the main result of the paper: if you want to mix snowcover models with avalanche dynamics simulations you must be very certain that the snowcover model is predicting the right fracture depth. Otherwise, simple model (VS) or more complicated models (wet snow) will provide the wrong results. That is, it is NOT about the flow model; good results can be obtained by both simple and more complicated procedures. This is our primary result. CHANGED: The introduction has been modified to make the goal of the paper more clear.**

- *Why does it make sense to use a mix of modeled snow cover parameters (depth) and guideline parameters?*

**ANSWER: In a "real" hazard mitigation analysis, the release depth is set by meteorological extremes. A specific avalanche is not modelled, but an event with a specific return period. Here we have a problem, which the reviewer has correctly identified: Our avalanche data base contains events with different return periods, most of them smaller than an extreme event. Frankly, we don't know the return period of our avalanches. To circumvent this problem we adopt the following approach: We take the modelled snow cover parameters (height, density) for the release, coupled with the entrained snow amounts, and use extreme friction parameters. We demonstrate that this approach CAN lead to good results. In fact, considering the procedure contains ONLY TWO friction parameters, and does not require detailed snowcover conditions, the results are surprisingly good! We apply this approach for all avalanches to make it general. Here we want the reader to come to the conclusion that maybe the guideline model is superiour to the more detailed wet snow avalanche calculation. Thus, our results indicate that good results, at small cost, can be achieved by mixing modeled snow cover parameters with guideline procedures.**

- *The growth indices should depend on the choice of flow and the entrainment model/parameters, so they are a result of the model chain?*

**ANSWER: Yes and no. Yes, the growth indices are clearly a result of the snowpack simulations which predict the snow distribution. They are therefore a result of the "model chain". The entrainment parameters do not vary strongly, but are limited to a small range. The growth indices are thus largely independent of the model parameters.**

- *With the thermomechanically modelled growth indices, the initial mass of the classical simulations are set. But since classical VS parameters where calibrated implicitly including entrainment (field observations that include entrainment) this should not be necessary?*

**ANSWER: The reviewer is correct. The classical model will provide the "same" results with or without the entrained mass. By including the entrain mass, however, we can argue that we have more "extreme" like events, and therefore can use extreme friction parameters which we apply. Again we find the procedure provides solid results at very low cost (i.e. less detail).**

- *Why is it appropriate to choose model parameters for small, frequent avalanches for all events, when release volume of e.g Gatschiefer is up to 330.000 m?*

ANSWER: Because we restrict ourselves to wet snow avalanches. For wet snow avalanches the friction paramters are (more-or-less) independent of size. This is a procedure often applied in practice. The reviewer is correct: we could not use the approach in general, especially for dry snow avalanches.

- *How can the ad hoc model parameter choices for the thermomechanical model be justified (that vary for different avalanche paths, e.g. Entrainment coefficient (0.6-0.8) and $\alpha$ parameter)? Or does the choice not matter because the result influence is negligible?*

ANSWER: They are very small variations. The influence on the results is neglible. In fact, we argue differently: a range of values provides very similar results and we cannot distinguish between specific values (e.g. simulations with entrainment coefficient 0.6 provided the same results as entrainment coefficient 0.8). Note that all other parameters remain constant for wet snow. Of course the exact quality of snow and terrain will vary, and thus the parameters. These parameters were varied for the avalanche path but once fixed were not varied in the permutations

*In section 3.5 ((ii) sensitivity analysis of the extended avalanche simulations with respect to avalanche path location) three different approaches to study the sensitivity of the thermomechanical model are performed (interchanging all or combinations of the model parameters that are related to the snow cover model - fracture and erosion depths, density, snow temperature and LWC). The sensitivity analysis is evaluated on a qualitative level, e.g. no single parameters ranges are investigated (varied with respect to their absolute values) with a quantification of the output variability (which would actually be the advantage of the introduced statistical measures). Open questions are:*

- *The snow cover model parameters are permuted by event location. With this no quantitative evaluation is possible with respect to the absolute variation of avalanche simulation input, which are (as depicted in table 2) $\approx 26\%$ for release depth, $\approx 16\%$ for densities and $\approx 46\%$ for LWC and $\approx 151\%$ for temperatures (compared to the respective mean value). Considering these differences (in magnitudes) a direct, systematic comparison and sensitivity analysis is hardly possible - how can we finally conclude which parameters are more important if the are not equally treated?*

CHANGED: We revised section 3.5 to address the comments of the reviewer. ANSWER: We think that the original manuscript failed to clearly describe our goal of the sensitivity study. One of the novel approaches we present is to use a physics-based snowcover model to determine the initial and boundary conditions for avalanche dynamics calculations. An important role of the sensitivity study is to determine if this approach add information in the simulation process. Here, we consider that all **12** cases represents a variety of wet snow avalanche cases, and the **12** simulations provide realistic, self-consistent initial conditions. Therefore, we decided to interchange the simulated initial conditions, instead to perturb the simulated values. For example, one could vary temperature over a range of -20 to 0 °C separately from LWC, but in this case, a well below freezing snowcover with a noticeable amount of liquid water is provided to the avalanche dynamics model. This is, however, not a realistic scenario. Therefore, we decided to interchange the sets of snowpack conditions from the SNOWPACK model.

*In section 4.3 ((iii) avalanche simulation sensitivity analysis with respect to computational/terrain resolution) the sensitivity with respect to grid size is evaluated. Main questions are:*

*As i understand it - this analysis treats the computational grid resolution. How is the DEM resolution treated (resampled to the computational resolution)?*

ANSWER: The measured DEM is resampled to the computational resolution. This is the standard procedure.

*The main result is that the presented method (statistical scores) can show that parameter values are bound to certain spatial resolutions. Since this has been observed before (e.g. by Bühler et al., 2011, as stated by the authors) this section could maybe be moved to the appendix to smooth the entire manuscript.*

ANSWER: The other reviewer also mentioned this point. We can consider removing the

section. However, we wanted to put the variation in our simulations that arise from interchanging the initial conditions in perspective. We show that the information added by using the SNOWPACK model is noticeable compared to changing calculation grid size resolution. Precisely because previous studies addressed already the influence of grid cell size on avalanche dynamics simulations, and researchers are aware of it, we consider it a good benchmark for the effect of initial conditions. We will revise the manuscript to make this more clear.

**Minor comments**

*Please find some more detailed line-by-line comments/questions below:*

- *title The title of the paper "Modeling the influence of snowcover temperature and water content on wet snow avalanche runout" could focus more on the main contributions (simulation evaluation/sensitivity) and results of the paper (as stated in the abstract Reliable estimates of avalanche mass (depth and density) in the release and erosion zones is identified to be more important than an exact specification of temperature and water content. - which slightly contradicts the title).*

**ANSWER: Valid point. It is about wet snow avalanche runout – but we find that it depends on the depth of the meltwater accumulation. I wonder if an alternative title could be, "The role of meltwater accumulation depth in wet snow avalanche modelling" or "Including snowpack properties in release areas for wet snow avalanche modelling". Is this a more appropriate title?**

- *abstract Do height and depth have different meanings? Is it consistent throughout the paper?*

**ANSWER: No, this is an inconsistency from our side. We will check for consistency when revising the manuscript. Because we denote the "depth" with "h" we will use height throughout.**

- *3, 66, . . . deposits area . . . deposition area prediction*

**ANSWER: Changed.**

- *3, 67, Instead of parameter optimization, . . . This is a crucial point. If you pursue a flow model comparison, both models should be equally treated, i.e. performing a full optimization and comparing the result performance, not to compare apples and oranges (c.f. Rauter et al., 2016, where a extended flow model is also compared to a Voellmy friction relation with different measures). If you pursue a comparison of simulation approaches/strategies, guidelines should not be mixed with model chain results.*

**ANSWER: Again our goal is not to "pursue" a flow model comparision. Both models appear to work well – if the snowcover model accurately predicts the meltwater accumulation zone. We hope that some readers will conclude that the VS approach is not too bad. Perhaps both models could be applied.**

- *3, 74, 3. . . . , Fig. 1 To me it appears that the "model chain" is the combination of snowpack and avalanche simulations. The statistical scores/analysis is valid tool to evaluate the results (jointly with the runout estimates) but not a part of the chain. Similar evaluations have been performed for operational avalanche simulations Naaim et al. (2013) (snow properties and simulated avalanche runout) or Fischer et al. (2015) and recently for other mass flows Mergili et al. (2017) (introducing statistical scores to evaluate model performance).*

**ANSWER: Yes we agree: different statistical methods could be used to evaluate the simulation results. We argue that some statistical procedure is necessary to compare the numerical results and therefore be included in the "model chain". We recognize that different models exist. However, we couple a three-dimensional avalanche dynamics model with a three-diemsional method to calculate statistical scores. This common component led us to include the statistical method in the modelling chain. We include the references.**

- *Section 2: Wet snow avalanche modeling In this section the underlying avalanche flow model is described. Since it corresponds to Valero et al. (2016) it could be omitted or transferred to the avalanche dynamic modeling (section 3.2 or appendix) part, as it distracts from the main topic of this paper.*

**ANSWER: We restructured this section so that it is easier to read.**

- *9, 219, . . . apply a three-dimensional avalanche dynamics model Maybe better: Two dimensional model operating in three dimensional terrain.*

**ANSWER: Changed.**

- *9, 221-223, The small elevation difference between the release zones and the weather stations . . . provides the sufficient conditions to . . . What do you mean with "sufficient conditions", i.e. sufficiently small?*

**ANSWER: We agree that the wording was not precise, we will formulate it as: "We argue that the elevation differences between the release zones or deposits zones and the weather stations (see Table 1) are sufficiently small to provide representative snowcover simulations to estimate the initial and boundary conditions of the case studies."**

- *13, 294, class Small avalanches. Same class for all release volumes from 4.000 $m^3$ up to 330.000 $m^3$ - is this in correspondence to (Salm et al., 1990)?. There are also reasons to assume that no mass/volume dependency is necessary and that parameters cannot be interchanged between locations (especially regarding non extreme events, c.f. Issler et al., 2005; Gauer et al., 2010).*

**ANSWER: We consider only wet snow avalanches. The procedure we apply cannot be used generally, that is, for dry avalanches. Because lubrication is the frictional mechanism driving wet snow avalanches, they are less dependent on size. (Unlike fluidization, which depends strongly on avalanche size, etc.)**

- *13, 298, Section 3.3 Contingency table analysis for deposition area How do these scores compare to similar approaches evaluating snow avalanche simulations (Fischer et al., 2015; Rauter et al., 2016) and other mass flow simulations Mergili et al. (2017)? Would it also be possible to show the result variability with only two of the scores (since they are based on two independent measures)?*

**ANSWER: In many ways we regard the methods of Fischer and Rauter to be superiour to the procedure adopted here. At least these methods consider other avalanche flow properties such as velocity. We simply can't apply these methods (and therefore make comparisons) because we are working on a set of documented case studies of wet snow avalanches. Thus, our method is simpler, but reduced. This does not mean that it should not be applied. Perhaps a hybrid method could be developed? But this is out of the scope of the paper.**

- *14, 316, section 3.4 Avalanche runout This is an interesting definition of runout in a simulation framework - what are the advantages and disadvantages of this definition (are there limitations for multipath effects?, c.f. Fischer, 2013)? Some more details on how the final time steps or simulation patterns are determined would be interesting (dependence on numerical parameters, e.g. cut off for flow depths? what are the stopping criteria/simulation times?, c.f. Teich et al. (2014)?).*

**ANSWER: Yes, the reviewer is correct. The two advantages of this approach are that (1) it is simple and (2) it is independent of the inundation area analysis. The disadvantage of this approach is that it does not consider the distribution of mass in the runout area (20cm cutoff etc.). The runout approach was designed to supplement the statistics of the inundation area hit, miss, false alarms. This statistical data can be misleading – therefore we think the combination of the two methods is appropriate. We will include information concerning the flow-depths and stopping criteria in the revised paper. We added information how we stopped the avalanches to section 3.4.**

- *14, 323, section 3.5 Influence of initial conditions on avalanche runout: sensitivity study and 17, 403, section 4.2 Sensitivity analysis The intention of an objective sensitivity analysis seems promising, but a systematic approach, which leads to clear and quantifiable results regarding the influence of single parameter/input variables is missing (see general questions above). The general result, that interchanging model parameters from one event to another, reduces the simulation performance is not surprising.*

**ANSWER: Section 3.5 has been modified. See also an earlier comment: the idea of exchanging event parameters is to maintain a consistent set of simulated snow covers. In the paper, we want to demonstrate that snowcover conditions that are required to drive the RAMMS-Extended model can be successfully derived from physics based snow cover models. In this regard, we were actually surprised that the connection between simulated snow cover conditions and the avalanche situation was so tight. One can argue that interchanging "true" initial conditions leads to the unsurprising result that the model performance reduces, but we consider it quite significant that this also holds for "simulated" initial conditions. In any case, we want to show that the simulations for an event indeed add information about the specific event. This motivated the exchange of snow cover conditions on an event basis. We think that our study showed that the snowpack model indeed contributes with accurate information about the snowcover conditions in the release area. In our opinion, a sensitivity study, as proposed by the reviewer, would address a different question, namely, purely focusing on the effect of single parameters. However, this approach would not guarantee consistent snowpack conditions. For example, varying the temperature while maintaining the liquid water content constant could lead to an non realistic condition of wet snow at temperatures well below freezing. So it may be considered a trivial result that event based snowpack conditions contribute to good model performance, but it is generally difficult to know the exact snowpack conditions of the release. Often it is dangerous to access release areas and particularly in wet snow avalanches, changes in the snow cover state can be very rapid, such that manual observations often miss the interesting period. Section 3.5 has been modified.**

- *16, 372, the guideline-VS model. The*

**ANSWER: Changed.**

- *25, 540, . . . such as speed, dynamic flow depths . . . . Is it possible to give an estimate on the magnitude of their variability?*

**ANSWER: In the discussion Sect. 5 we eplicitely state, "Other important avalanche variables, such as speed, dynamic flow heights and impact pressures are not considered in the analysis, although they are crucial in many aspects of assessing avalanche risks." We have compared our model to available velocity data. However, this data is often restricted to specific test sites (e.g. VdlS) and is often incomplete, in the sense that snowcover data is missing. In our paper we attempt to model the interaction between snow AND terrain and therefore believe it is better to have different terrain, although the velocity data is missing.**

- *25, 540-542, . . . avalanche risks. What would be the benefit of using further modeling results? Why is it not necessary to consider them ( compare, e.g. for avalanche velocities Sailer et al. (2002); Ancey and Meunier (2004); Gauer (2014) or Sovilla et al. (2007); Fischer et al. (2015) for growth indices/mass balance)?*

**ANSWER: If you have the data then one MUST use them (speed, entrainment). However, this is not the usual case. We would like to turn the problem around: how can you best use a massive amount of data (inundation areas) to the greatest possible advanatage. We understand that the analysis is not complete. But we are considering a sub-class of avalanches (wet snow avalanches) where even velocity data is sparse. Data obtained from test sites is likewiselimited because it contains only one terrain geometry, or overlapping avalanche events.**

- *26, figure 11 For better comparability the same scaling of the y-axis of the single figures (a), (b) and (c) would be desirable.*

**ANSWER: Thanks for the suggestion. The figures have been modified accordingly.**

- *28, 635, . . . depth and spatial extent of the avalanche release area was known. How does the SNOWPACK model perform regarding the documented release depths - are there any measurements available?*

**ANSWER: We cite five SNOWPACK papers concerning model validation in the text. Three of them were written by one of the co-authors and are related to wet snow modelling using field data from the area where seven of our case studies occured. We consider that the snow cover model was tested enough and we do not consider this manuscript as the place to re-evaluate the snow cover model performance. Only for some case studies are there fracture depth measurements performed with a laser scan, see Table 1.**

**Response to review by G. Chambon**

*I commend the authors for the impressive amount of work summarized in this paper: the compilation of data, systematic SNOWPACK and RAMMS simulations, and extensive sensitivity study provide an unprecedented set of results concerning the modelling of wet snow avalanches and the influence of various parameters such as initial mass and snow temperature / LWC on avalanche deposits and runouts. Despite the complex chain of models that is used, the authors made the effort to try to isolate the most influential physical processes, which I find particularly interesting. I am henceforth fully favorable to the publication of this paper in NHESS. I think however that several aspects of the paper could be improved to provide a better account of this nice study. First, the paper is a bit lengthy and redundant at places, and the structure of certain sections could be improved. Most importantly, I feel that the choice made by the authors to base most of the discussions on the statistical scores coming from the contingency table analysis, sometimes tend to "soften" the results and "dilute" the differences among the models. Putting more emphasis on more physical outputs, such as the raw results shown in supplementary material and the runout distances, would help counterbalance this trend. Finally, I consider that the discussion of the sensitivity analysis needs to be complemented with more quantitative comparisons and discussions. The specific comments below provide more detailed suggestions on these issues.*

**ANSWER: We thank G. Chambon for his positive judgment of our work as well as the constructive comments. Please find a detailed response to the issues raised by him below.**

**Specific comments**

*1/The introduction would benefit from being more to the point at certain places. The second paragraph, in particular, appears a bit off-topic and overly speculative. If the goal is to explain that wet snow avalanches are characterized by relatively large values of apparent viscosity and cohesion, there is probably no need to discuss the so-called "compactive strength" of snow and its hypothetical relation with viscosity. On the other hand, in the third paragraph, a more in-depth discussion of the advantages and drawbacks of the different approaches used in past studies to model wet snow avalanches would be in order.*

- **ANSWER: We agree with you. We removed the speculative part of paragraph. This makes the text clearer and less redundant. Please see the track changes file**

*2/Section 2, presentation of the model: A clearer structure (e.g., avoiding redundancies and introducing subsections / subtitles to better distinguish between the different elements of the model) would improve the readability of the section. Moreover, certain mathematical notations could probably be simplified, and some physical relations better explained.*

**ANSWER: We broke up the section into different subsections and removed the all the redundancies we could find. Please see the track changes file**

Some suggestions below:

- Why using the subscript $\Phi$ everywhere? Is it really useful?
  **ANSWER: In order to make this work consistent with previous works it is important to keep the $\Phi$. All variables subscripted with $\Phi$ refer to the avalanche dense/flowing part.**

- The variable $N_K$ present in Eq.(1) would need to be defined earlier after this.
  **ANSWER: Yes, you are right. We define $N_K$ earlier. Line 114 of the new manuscript**

- What is the parameter $\gamma$ in Eq. (3)?
  **ANSWER: Yes, you are right. We removed the $\gamma$ from the Eq. 3 (Eq. 7 in the new manuscript) and define the parameter $\gamma$ in the lines inmediately below. Thank you.**

- What are the quantities $h_\Phi s$ and $\rho_\Phi s$ in line 117?
  **ANSWER: These variables represent the co-volume height and density. The co-volume represents the densest possible packing of snow granules in the avalanche core. We don't want to talk about this too much because it goes into too much detail, so we simply placed a citation in the text. See the track changes file**

- Indicate the physical meaning of $S_\Phi$ (shear stress).
  **ANSWER: We now write "The shearing stress ..." We have created an entire section entitled "Flow friction" where the shear stress is described in detail. See the track changes file**

- The sentence starting with "The basal boundary converts ..." on line 129 is not very clear. This point would maybe be better explained in conjunction with Eq. (7)?
  **ANSWER: yes, you are correct, we placed this text after Eq. 7.**

- What is the relation between the quantities $\dot{P}_\Phi$ and $R_\Phi$? Why not denoting the former simply as $\dot{R}_\Phi$.
  **ANSWER: The variable $P$ denotes the input of energy (source term) whereas the variable $R$ denotes the value of energy after ALL processes (advection, sinks) have been considered. We too would like them to be the same, but this is mathematically impossible. Commented in the lines 144-159 in the new manuscript**

- Idem: what is the relation between $\dot{P}_\Phi V$ and $R_\Phi V$?
  **ANSWER: Please see above. Lines 144-159 new manuscript**

- What is the coefficient $c$ in Eq. (11)?
  **ANSWER: Corrected. It is the specific heat. The subscript $\Phi$ is missing in the equation. Eq. 11 in the new manuscript**

- The sentence starting with "Equation (14) takes into account..." in line 174 is not very clear.
  **ANSWER: We now write, "Equation ... takes into account the thermal energy contained in the entrained snow." This is better, because we avoid the use of the word "production" which confuses everything. See track changes file**

- The specific form chosen for the cohesion, i.e. the factor $(1 - \mu)$ and the exponential term, should be commented.
  **ANSWER: This specific form of the cohesion function is based on results from snow chute experiments. These experiments show that the shear stress increases from zero $S_\phi = 0$ when the normal stress is zero $N=0$. Basically the form of this function comes from fitting measurements. In the text we write, "The form of Eq. 16 ensures that the shear stress $S_\mu=0$ when $N=0$, in accordance with shear and normal force measurements in snow chute experiments."**

*3/Section 3.1. It is not fully clear whether SNOWPACK simulations were performed only for the release zones, or also for the deposition zones (in cases where data are available for these zones).*
**ANSWER: SNOWPACK simulations were also performed when a station in the valley was available (9 out of 12 cases). This is shown in Table 1. For the valley simulations, the virtual slopes were not considered, and only the flat field simulations were used. This corresponds**

to the fact that deposits area for large avalanches are relatively flat, compared to the release area. See line **253-265** from th new manuscript.

*4/Section 3.1, Table 3. How is the erodibility coefficient obtained? This parameter is not discussed in the text, although its influence on the results is probably far from negligible.*

**ANSWER: Yes, the reviewer is correct. We selected the erodibility coefficient based on extensive back-calculation of wet snow avalanche events. The selection process is reported in a previous paper. We don't want to clutter up the paper here, but we introduced the sentence in section on entrainment: "The value of the erodibility coefficient depends on snow quality. Values for warm, wet snow are reported in Vera et al. (2015, 2016).". See track changes file.**

*5/Section 3.2. The value chosen for the parameter $\zeta$ involved in Eq. (7) should also be discussed.*

**ANSWER: we made a notation mistake here. The parameter $\zeta$ does not exist, it should be $\gamma$. We also write, "The fluidization parameters $\alpha$ and $\gamma$ (please see Bartelt et al. (2006) and Vera et al. (2016)), are fixed to a pre-determined values based on the terrain characteristics for each avalanche path. Once these parameters are fixed they are not tuned for the remaining set of simulations.". See the track changes file.**

*6/Section 3.2. Besides data on avalanche release area, the authors probably also have data on fracture depths for at least some of the avalanches. How do these data compare to the fracture depths predicted by SNOWPACK? Where they used in way to optimize the results of SNOWPACK?*

**ANSWER:There are fracture depth data in the avalanche measured with laser scan and drone. The fracture depths measured are obviously not constant but taking avergae values are in good agreement with the values estimated by SNOWPACK, see Table 1. The reported fracture depth data was used to constrain the SNOWPACK simulaitons. This data was therefore very helpful in determing the quality of the SNOWPACK simulations.**

*7/Section 3.2. Regarding the Voellmy-Salm model, and if I understood well, the authors chose to use the same friction parameters for all studied avalanches. Would not it make more sense to optimize these parameters for each avalanche? I do not see any reason why all these avalanches should be characterized by identical friction parameters. In addition, giving the value of these parameters would also be useful, for the sake of comparison with the parameters used in the RAMMS model.*

**ANSWER: We did not change the parameters of the thermomechanical model – they were fixed to values. We tried to run the thermomechanical model, changing only the initial and boundary conditions. We wanted to do the same for the Voellmy model. It is extremely important to us that WE DO NOT optimize the friction parameters for a particular avalanche – either for the thermomechanical model or the Voellmy. We have a set of "wet snow parameters" that we use for all wet snow avalanches. The initial (release) and boundary conditions (terrain, snowcover) are changed for each avalanche. We emphasize this result in the conclusions. In practice a user of the Voellmy model will follow the same approach – they will use the guideline parameters. We are not comparing models: we are comparing two approaches: A thermomechanical modelling approach where we change only the initial and boundary conditions against the standard Voellmy model.**

*8/Section 3.5 is not very clear and some redundancies could be avoided. In the first paragraph, in particular, it is difficult to understand what the 432 simulations represent, whereas this issue is better explained afterwards.*

**ANSWER: We rephrased this section completely, also to address the comment by the other reviewer that the motivation for the way the sensitivity study was set-up was not clearly explained. See the track changes file.**

*9/Section 4.1. While the different statistical scores used by the authors effectively show that the thermomechanical model performs better than the Voellmy-Salm model, this issue is even more evident from observation of the model outputs provided as supplementary material. Hence I would encourage the authors to add, at least, a short description of these raw outputs in the main text prior to discussing the statistical scores. Adding a figure showing one or two illustrative examples of raw results in the main text could also be option. Similarly, moving the runout comparisons (currently presented in 4.4) before the statistical score comparisons could also help to better illustrate the differences among the models.*

**ANSWER: We agree. We inserted the following text before we begin to discuss the statistical scores: "The results of the model runs are presented extensively in the paper supplements.**

The graphs in the supplement A facilitate a direct comparison between the thermomechanical approach, the standard Voellmy-Salm procedure and the actual avalanche measurements, including the location of the deposits with respect to the observed release zone. Supplement B contains the results of the model permutations. This graphical output enables a quick assessment of the model sensitivity. In the following we statistically analyze model performance."

*10/Section 4.1. The sentence starting by "The fact that the difference in ETS score ... "in line 379 seems in contradiction with what is said just before (lower difference in ETS than in HKS between the two models).* **ANSWER: You are correct, it's in contradiction. The sentence was wrong and referred to an earlier version of the graph. We removed it from the manuscript. The difference in POD between the thermodynamics model and Voellmy-Salm model is larger than the FAR. So in the results presented, the Hanssen-Kuiper skill score is not biased towards the Voellmy-Salm model anymore, as the POD is not higher.See the track changes manuscript**

*11/Section 4.2, line 407. Why do the authors refer to the friction parameters used in the VS model as "extreme" here?* **ANSWER: extreme refers to avalanche with return periods greater than 300 years. We now state in the text, "The primary result of the preceding section is that guideline-based avalanche dynamics models with extreme friction parameters (avalanches with return periods greater than 300 years) will have difficulty reconstructing individual case studies and that they are not easily linked to snowcover conditions."**

*12/Section 4.2-4.3-4.4. I encourage the authors to provide more quantitative evidences of the conclusions drawn from their sensitivity study. In the current manuscript, it is sometimes difficult to relate the assertions made in the text to the presented data. One probable reason is that the authors rely throughout on the same type of figures, whereas alternative representations, such as boxplots or distributions / percentiles, would probably allow for easier quantitative comparisons between, e.g., the different initial conditions (mass versus temperature/LWC) or the different grid resolutions. I indicate below a few examples of overly qualitative statements that would need to be supported by more quantitative evidences:*

- *line 426: "generally higher"* **CHANGED: removed "generally". They are higher, see graphs 5 and 6.**

- *line 431: "the simulation with the original initial condition is among ..."* **CHANGED: removed. Adds no additional information.**

- *line 440: "are more sensitive to"* **CHANGED: removed. Adds no additional information.**

- *line 452-453: A small variation (...) would lead to a large variability (While Fig. 6 shows that simulations with other initial conditions are sometimes as good as simulations with the correct initial conditions.)* **CHANGED: We now quantify small. A change in the fracture depth of 10cm can lead to a large variability in the predicted avalanche runout. This is a problematic result because it indicates the critical role of fracture depth as an input parameter in avalanche simulations.**

- *line 459: "less sensitive"* **Removed and shortened the text.**

- *line 465: "The variation was strongest"* **CHANGED: We write: The strong variation on long avalanche tracks with a smooth transition to runout zone demonstrates, once again, that path geometry dominates over changes in snowcover boundary conditions.**

*13/Section 4.2.2: Could the authors also discuss the relative influence on the results of mass in the release area versus mass in the entrainment zone?* **ANSWER: The statistical scores show superior scores when the correct entrainment conditions are modelled. However, The results are controlled by the water content/warmth of the entrained snow. The problem is, and we have stated this in the work, that the water content/warmth of the entrained snow did not vary strongly, because we are considering only wet avalanches. The role of entrainment would change dramatically, if we were to include dry and wet snow avalanches. We have added the text, "The role of mass entrainment is difficult to**

identify in the statistical scores because we considered only warm/moist snowcovers. Moreover, the permuations did not include dry, cold snowcovers. This result suggests that the snow quality (temperature, moisture) is more important than the snow amount.". see track changes manuscript.

*14/Section 4.3: The description of the effect of grid size on the statistical scores could probably be shortened, and redundancies avoided. I suggest however to extend the – currently very short – last paragraph describing the interplay between initial conditions and resolution. To me, this latter issue constitutes the real novelty of the sensitivity study conducted by the authors with respect to grid resolution.*

**ANSWER: The reviewer is correct. The very short paragraph should not stand alone as it begs for further detail and explanation. We deleted it from the results. The contents are covered in the discussion section.**

*15/Section 5: The sentence starting with "Moreover, the connection between friction and initial starting mass" in line 597 is not very clear.*

**ANSWER: Yes, we agree. It adds no further information. We deleted it from the paper.**

*Table 1. The caption mentions virtual slope, but this information does not seem to appear in the table?*

**ANSWER: The third column in the table should have been interpreted as, for example, KLO3-NE means AWS is KLO3, virtual slope is North-East (NE). Changed the table caption and table layout to make this more clear.This info is in table 2 too.See track changes file.**

*Line 249-252: The sentence starting with "In case of avalanches with new snow …" is not fully clear: does it apply only to the cases where meteorological data in the deposition zone are not available, or to all cases?*

**ANSWER: This indeed is not clear. We only use data from the measurment stations, which are located in the release zones. No data is really available for the deposition zones, which are based on snowcover modelling. Therefore, it applies to ALL cases.**

*Line 308: The reference to Table 2 seems wrong here.*

**ANSWER: Thank you for pointing out, it should have been Fig. 2 instead of Table 2.**

*Fig. 3, caption: word missing after "the longest calculated".*

**ANSWER: Thank you for pointing out, changed to: "the longest calculated flowline (red dot)"**

*line 477: typo: "courser"*

**ANSWER: Thank you for pointing out, changed to "coarser".**

*Fig. 10: Why the asterisk with the specific value corresponding to the CV-1 case?*

**ANSWER: For this case study we had a 1m digital elevation model, obtained from a drone flight. We added in the caption: "It was necessary to simulate the CV-1 case with a 1m grid resolution to better account for a vertical wall."**

*line 524: why the "(not shown)", instead of a reference to section 4.3 where variations of ETS and HKS with resolution are extensively discussed?*

**ANSWER: We removed "not shown" Made referene to section 4.3.**

*Fig. 11: (a), (b), (c) need to be added to the plots.*

**ANSWER: Thank you for pointing out, the figure was corrected.**

*line 609: word missing after "the maximum LWC"?*

[revised manuscript text omitted]
 (%) upper | LWC (%) lower | Erosion depth (m) upper | Erosion depth (m) lower | Erosion depth gradient (m/100m) upper | Erosion depth gradient (m/100m) lower | density (kg/m$^3$) upper | density (kg/m$^3$) lower | volwater (mm/m) upper | volwater (mm/m) lower | temperature (°C) upper | temperature (°C) lower | temperature gradient (°C/100m) upper | temperature gradient (°C/100m) lower | erodibility (-) upper | erodibility (-) lower |
|---|---|---|---|---|---|---|---|---|---|---|---|---|---|---|---|---|
| Gruenbodeli | 1.45 | - | 0.56 | 0.00 | 0.02 | - | 197 | - | 8.1 | - | -0.2 | - | 0.0 | - | 0.8 | - |
| Salezer | 1.89 | - | 0.95 | 0.00 | 0.03 | - | 317 | - | 18.0 | - | 0.0 | - | 0.0 | - | 0.7 | - |
| Gatschiefer | 0.00 | 1.47 | 0.55 | 0.95 | 0.03 | 0.04 | 185 | 360 | 0.0 | 14.0 | -1.0 | 0.0 | 0.0 | 0.0 | 0.6 | 0.7 |
| Braemabuhl 2013 | 2.97 | - | 1.11 | 0.00 | 0.04 | - | 353 | - | 33.0 | - | 0.0 | - | 0.0 | - | 0.6 | - |
| Drusatscha | 3.41 | - | 0.54 | 0.00 | 0.02 | - | 291 | - | 18.4 | - | 0.0 | - | 0.0 | - | 0.6 | - |
| MO-4 Andina Chile | 2.44 | - | 0.90 | 0.00 | 0.03 | - | 296 | - | 22.0 | - | 0.0 | - | 0.0 | - | 0.6 | - |
| Grengiols | 0.00 | 4.67 | 0.43 | 0.60 | 0.03 | 0.00 | 175 | 270 | 0.0 | 28.0 | -7.4 | 0.0 | 1.5 | 0.0 | 0.7 | 0.8 |
| Verbier Mont Rogneux | 3.00 | - | 0.60 | 0.00 | 0.02 | - | 317 | - | 18.0 | - | 0.0 | - | 0.0 | - | 0.6 | - |
| Verbier Ba Combe | 2.59 | - | 0.58 | 0.00 | 0.02 | - | 349 | - | 15.0 | - | 0.0 | - | 0.0 | - | 0.6 | - |
| Braemabuhl verbauung | 0.00 | 1.41 | 0.25 | 0.85 | 0.00 | 0.04 | 158 | 335 | 0.0 | 12.0 | -2.0 | 0.0 | 0.0 | 0.0 | 0.8 | 0.8 |
| Braemabuhl Wildi | 0.00 | 1.25 | 0.30 | 0.80 | 0.00 | 0.03 | 164 | 335 | 0.0 | 10.0 | -2.0 | 0.0 | 0.0 | 0.0 | 0.6 | 0.6 |
| CV-1 Andina Chile | 1.51 | - | 0.37 | 0.00 | 0.00 | - | 359 | - | 5.6 | - | -0.1 | - | 0.0 | - | 0.6 | - |

[revised manuscript text omitted]

---

## Referee Report (RR1)

**Review of 'Modeling the influence of snowcover temperature and water content on wet snow avalanche runout' by C. VERA VALERO et al.'**

In their revision the authors provided detailed answers to the issues raised by the reviewers and managed to submit a new, enhanced version of their manuscript. The updated structure allows to follow the ideas of the authors in a better way, but some issues remain a bit unclear (especially section 4.1). Some clarifications that would especially be helpful for the potential readers, are addressed in the answer to the reviewers, but are missing the in the updated manuscript and could be considered before publication. In the following i summarize some main points; page and line numbers refer to the updated version of the manuscript with highlighted changes in red and blue:

- depth and height are still used inconsistently throughout the manuscript

- *3, 70: typo: citepGruber2009*: I am not sure this is completely true, see comment below (parameter $\alpha$)

- *9, 225*: Here, it would be worth to mention whether the the computational resolution always $3\text{x}3\,\text{m}$ (besides the resolution variations in section 4.3)?

- *10, 245 ...are sufficiently small...*: Why are they sufficiently small?

- *10, table 1*: I completely agree that this manuscript should not focus on any detailed evaluation of SNOWPACK, however given the observed fractures depths (table 1), it would be very interesting how it compares to the derived release depth - since it is a crucial part of the analysis.

- *15, 340: ..This two-dimensional procedures avoids the problem of defining a one-dimensional measure of avalanche runout.* I completely agree with the authors: However since you already spend the effort of calculating it, it could (but it does not need to) also be included in the analysis? Furthermore it would still be interesting how these evaluation measures compare to other measures used for avalanche simulations evaluation (such as mentioned in the first review, e.g. (Mergili et al., 2017a, for 2d) or (Teich et al., 2013, and refs therein) for definitions of runout).

- *deposition*: For my point of view it is completely appropriate (as done and stated by the authors) to evaluate simulation results of flow depths (above a threshold) at the last time with deposition patterns, without taking into account different, observed depositions depths. However for the purpose of clarity, it should be stated that (i) *deposition* is an interpretation of this distinct simulation result, since the model does not directly cover deposition mechanisms (see e.g. Mangeney-Castelnau et al., 2003; Mergili et al., 2017b) and (ii) how this simulation result is defined numerically (i.e. *15, 352: ...all simulations stopped when 95 percent of the total mass stopped moving* - does the 95% correspond the maximum momentum? what does stop mean (velocity below certain threshold?) in this context?)

- *section 4.1:* : I have no doubt that the thermomechanical model allows to gain different and probably even more suitable simulation results that the classical approach - but i am not convinced that the approach used here is valid to draw this conclusion *18 430: ...the thermomechanical model statistically outperforms the guideline procedure...*: While ad hoc parameter assumptions are *allowed* for the thermomechanical model (*The model has one parameter $\alpha$ ... chosen by the avalanche expert...*) - guideline values (that still seem a bit unclear to me (for *wet snow avalanches* or *13, 323 return period of 10 or 30 years...* or *19, 459: return periods greater than 300 years*)) are used for the guideline-VS model. Also considering that release depth is the most sensitive parameter, it could be discussed in more detail why it is appropriate to perform simulations with the *thermomechanically ad hoc modelled* total mass (release and entrainment) with the vs-guideline model, see *lines 298– 323: ...include the entire avalanche mass within the release volume...* - which would yield that the *...total mass in both simulations is similar...* but as it could also be seen: a different setup in initial

potential energy of the flow. That said, it appears to me that the comparison (more specifically this section) is not performed in a *fair* way.

- *section 4.3:* It could be interesting to shortly comment on how the DEM resolution (or computational/numerical resolution) may have an influence on the release volume (in terms of release area etc.) or not since one main outcome of the paper is that very small variations of release volume have a large impact on the simulation results (*28, 624: an underestimation of fracture depth of only 10 cm could lead to significant runout shortening*).

**References**

Mangeney-Castelnau, A., Vilotte, J.-P., Bristeau, M., Perthame, B., Bouchut, F., Simeoni, C., and Yerneni, S. (2003). Numerical modeling of avalanches based on Saint Venant equations using a kinetic scheme. *Journal of Geophysical Research*, 108(B11):2527.

Mergili, M., Emmer, A., Juicov, A., Cochachin, A., Fischer, J.-T., Huggel, C., and Pudasaini, S. P. (2017a). How well can we simulate complex hydro-geomorphic process chains? the 2012 multi-lake outburst flood in the santa cruz valley (cordillera blanca, per). *Earth Surface Processes and Landforms*, pages n/a–n/a. ESP-16-0360.R3.

Mergili, M., Fischer, J.-T., Krenn, J., and Pudasaini, S. P. (2017b). r. avaflow v1, an advanced open-source computational framework for the propagation and interaction of two-phase mass flows. *Geoscientific Model Development*, 10(2):553–569.

Teich, M., Fischer, J.-T., Feistl, T., Bartelt, P., Bebi, P., Christen, M., and Grêt-Regamey, A. (2013). Evaluation and operationalization of a forest detrainment modeling approach for computational snow avalanche simulation. In *AGU General Assembly 2013, San Fransisco, USA*.

---

## Author Response (AR2)

**Response to review by J.-T. Fischer**

*Review of 'Modeling the inuence of snowcover temperature and water content on wet snow avalanche runout' by C. VERA VALERO et al.' In their revision the authors provided detailed answers to the issues raised by the reviewers and managed to submit a new, enhanced version of their manuscript. The updated structure allows to follow the ideas of the authors in a better way, but some issues remain a bit unclear (especially section 4.1). Some clarifications that would especially be helpful for the potential readers, are addressed in the answer to the reviewers, but are missing the in the updated manuscript and could be considered before publication. :*

**ANSWER: We thank the reviewer again for the detailed comments. The manuscript is continually improving.**

*In the following I summarize some main points; page and line numbers refer to the updated version of the manuscript with highlighted changes in red and blue:*

- *depth and height are still used inconsistently throughout the manuscript*

**ANSWER: We are sorry. We went through the manuscript and consistently write "height" instead of "depth", where appropriate**

- *P3, 70: typo: citepGruber2009 : I am not sure this is completely true, see comment below $\alpha$ (parameter)*

**ANSWER: Changed, thank you, see track changes manuscript.**

- *P9, 225 : Here, it would be worth to mention whether the computational resolution always 3x3m (besides the resolution variations in section 4.3)??*

**ANSWER: Thank you for the suggestion. We mention now the used grid resolutions in section 3.2. Note that we think that Section 3.2 is more appropriate than the proposed Section position by the reviewer.**

- *P10, 245 ...are sufficiently small...: Why are they sufficiently small?*

**ANSWER: The elevation differences between the weather stations and the release areas are typically less than 200m. When we compare this to typical lapse rates in the atmosphere (e.g., 7 degrees / 1000m for air temperature), we consider them representative for the release areas, given typical uncertainties in weather station measurements. We rephrased the manuscript at this point. We write, The elevation differences between the release zones or deposits zones and the weather stations are typically less than 200m, which we consider sufficiently small, given typical lapse rates in the atmosphere, to provide representative snowcover simulations to estimate the initial and boundary conditions of the case studies**

- *P10, table 1 : I completely agree that this manuscript should not focus on any detailed evaluation of SNOWPACK, however given the observed fractures depths (table 1), it would be very interesting how it compares to the derived release depth - since it is a crucial part of the analysis.*

**ANSWER:Note that Table 1 lists for several cases the release depth derived from measurements using either a drone, or terrestrial laser scanning. These were unfortunately replaced by "xx" in the track changed version, so the reviewer may have missed it. However, we added a paragraph explicitly comparing the field measurements with the SNOWPACK simulations results. See the track changes manuscript**

- *P15, 340: ..This two-dimensional procedures avoids the problem of determining a one-dimensional measure of avalanche runout. I completely agree with the authors: However since you already spend the effort of calculating it, it could (but it does not need to) also be included in the analysis? Furthermore it would still be interesting how these evaluation measures compare to other measures used for avalanche simulations evaluation (such as mentioned in the first review, e.g. (Mergili et al.,2017a, for 2d) or (Teich et al., 2013, and refs therein) for definitions of runout.*

**ANSWER:We use both the 2d contingency and the 1d runout analysis.**

- *deposition: For my point of view it is completely appropriate (as done and stated by the authors) to evaluate simulation results of flow depths (above a threshold) at the last time with deposition patterns, without taking into account different, observed depositions depths. However for the purpose of clarity, it should be stated that (i) deposition is an interpretation of this distinct simulation result, since the model does not directly cover deposition mechanisms (see e.g. Mangeney-Castelnau et al., 2003; Mergili et al., 2017b) and (ii) how this simulation result is defined numerically (i.e. 15,352: ...all simulations stopped when 95 percent of the total mass stopped moving - does the 95% correspond the maximum momentum? what does stop mean (velocity below certain threshold?) in this context?)*

**ANSWER: We write: 'The calculated flow height at the last calculation step, provides us with the inundation area. These flow heights might not represent the observed deposition depth, which is governed by different deposition mechanisms. The correspondence of observed and calculated inundation area is checked using a dichotomous contingency table' We rephrased the stopping criterion: 'All simulations stopped when the avalanche simulation contained less than 5% of the maximum calculated momentum'.**

- *section 4.1: I have no doubt that the thermomechanical model allows to gain different and probably even more suitable simulation results that the classical approach - but i am not convinced that the approach used here is valid to draw this conclusion P18 430: ...the thermomechanical model statistically outperforms the guideline procedure...: While ad hoc parameter assumptions are allowed for the thermomechanical model (The model has one parameter - ... chosen by the avalanche expert...) - guideline values (that still seem a bit unclear to me (for wet snow avalanches or 13, 323 return period of 10 or 30 years... or 19, 459: return periods greater than 300 years)) are used for the guideline-VS model. Also considering that release depth is the most sensitive parameter, it could be discussed in more detail why it is appropriate to perform simulations with the thermomechanically ad hoc modelled total mass (release and entrainment) with the vs-guideline model, see lines 298 to 323: ...include the entire avalanche mass within the release volume... - which would yield that the total mass in both simulations is similar... but as it could also be seen: a different setup in initial potential energy of the flow. That said, it appears to me that the comparison (more specifically this section) is not performed in a fair way.*

**ANSWER: We want to emphasize to the reviewer: We do not want to compare models. Independent of the model and the model results, we identify problems. Clearly a thermomechanical model, that accounts for snow temperature and snow conditions must be applied. We truthfully characterize the problems of model applications for frequent avalanche within the framework of a real time type hazard mapping. Nowhere in the conclusions do we state that the thermodynamical model should be applied, or, that the guidelines should be replaced. We simply point out the problems of specifying initial conditions based on the best method possible, snowcover simulations.**

**In the discussion we write: 'The general thermomechanical avalanche dynamics model RAMMS performs better than the guideline-VS model in all statistical scores, HKS, ETS, POD and FAR. The guideline procedures are designed to model extreme, dry flowing avalanches, not particular avalanche events. However, the guideline model achieved in some cases high**

contingency table scores, despite the application on non-extreme, wet snow avalanches. The guideline-VS model was forced using friction coefficients calibrated by Salm 1990. It was necessary to use the friction coefficients corresponding to smaller avalanche sizes in order to achieve a good correspondence between measurements and simulations. For all case studies, the friction coefficients chosen correspond to size class 'Small' and a return period of 10 to 30 years. The guideline-VS model had to be manipulated by an expert user to get the best results. For example, the general model was first applied to determine the mass-balance of the event, which was then used to establish the initial conditions (i.e., released plus eroded mass) of the guideline-VS model. Another disadvantage of the guideline model is that first a calibration of the friction parameters is required to obtain reasonable contingency table scores. Both steps are not required in the general model applications, because the friction parameters are determined as a known function of snowcover conditions.'

- *section 4.3: It could be interesting to shortly comment on how the DEM resolution (or computational/numerical resolution) may have an influence on the release volume (in terms of release area etc.) or not since one main outcome of the paper is that very small variations of release volume have a large impact on the simulation results (28, 624: an underestimation of fracture depth of only 10 cm could lead to significant runout shortening*

**ANSWER: In the summary of the DEM resolution section we make a summary of the DEM resolution analysis. It should be clarify now.**

**Response to review by G. Chambon**

- *P.2, l.24-29. I still do not understand the relations implied between the low cohesive strength of wet snow and the size of the snow granules, and between the size of the granules and the effective viscosity and cohesion of the flow. Please consider adding further information / references on these issues, or simplifying these statements. From what I understand, the important message here is that wet snow avalanches tend to have larger viscosities and cohesion; speculative and questionable interpretations on the microstructural origin of these trends are probably not necessary.*

**ANSWER: We simplified the text. We now write 'The runout of wet snow avalanches is especially difficult to calculate because temperature and liquid water content (LWC) have a strong influence on the mechanical properties of snow Denoth 1982, Voytoyvski 1977, Salm 1982. When warm snow contains liquid water, the deformation mechanics is controlled by the liquid film at the grain to grain contact Salm 1982. Wet snow can be plastically deformed until it reaches "packed density". Granules in wet snow avalanches are therefore large, heavy and poorly sorted in comparison to granules in dry avalanches. The bulk flow viscosity and cohesion of wet snow avalanches is larger than in dry flows. The formation of levees with steep vertical shear planes in wet snow avalanche deposits is another indication of the viscous and cohesive character of wet snow avalanche.'**

- *P.7, l.136-139. It is not clear how the newly-added sentences starting with The basal boundary connect with Eq. (7). Does it mean that Eq. (7) only account for processes active at the base? I would have thought that shear-induced dilation or compaction can also occur in the bulk of the flow (as in granular materials)*

**ANSWER: Yes, you are correct. The shear induced dilation can also occur in the bulk of the flow. However, the model uses depth-averaged approach, therefore we cannot separate processes in the core or at the base. Here we state that a hard rigid boundary is necessary to**

dilate the core and to initiate the change in configuration. We now write, The basal boundary converts the production of random kinetic energy $\dot{P}_\Phi$ in the bulk into an energy flux that changes the $z$-location of particles and therefore the potential energy and particle configuration of the core.

- *P.12. Is the amount of erodible snow determined similarly to the initial fracture depth, i.e. based on the location of highest LWC in the snowpack? If so, this information would need to be stated more clearly in the text. Moreover, recalling again, at the end of this paragraph, how the erodibility coefficient were obtained, would certainly be useful.*

**ANSWER: When analyzing the SNOWPACK simulation results, we found that the snow cover stratigraphy at the valley bottom often bears little similiarity to the snow cover in the release area. In the valley, multiple melting events may have occurred during the winter season, homogenizing the snowpack. In homogeneous snowpacks, ponding occurs less often. Also crusts that formed earlier in the winter season in the valley may cause ponding at a depth that does not correspond to the fracture depth in the release area.**
**We now write, 'The amount of erodible snow is also calculating using the location of the ponding layer. However, we calculate a gradient between the snowcover conditions at the release and the conditions at the valley bottom. This means that the depth of the fracture height and erodible layer decrease with elevation. The erosion model used is described by Christen 2010, Bartelt 2012.' We think this is much clearer than before. The erodibility values are the same as stated in the reference papers. All simulations were performed within a narrow band of 0.6 and 0.8.**

*P.14-15. The newly added sentence - 'This two-dimensional procedure avoids the problem of defining a one-dimensional measure of avalanche runout ' - appears at odd with the next paragraph, which specifically deals with the definition of avalanche runout*

**ANSWER: We introduced this sentence to explain an issue noted by reviewer no 1. This line explains why we use a contingency table analysis (2D) and a runout analysis (1d). Thus, we perform both a 1d and 2d analysis. If you use 'runout' you need to define exactly what runout is. In the contingency table analysis, no definition is necessary.**

*P.18, l.419. I do not understand why the authors refer here to extreme avalanches with a return period greater than 300 years, while it is said previously (p.13, l.298-299) that the Voellmy-Salm parameters used correspond to the class of small avalanches with return periods of 10-30 years?*
**ANSWER: Yes. this is clearly a mistake we write '(avalanches with return periods greater than 10 years)'.**

*P.19, l.444-445. The sentence 'The average scores of all' seems fully redundant with the one just 4 lines before: 'The average of the four'.*

**ANSWER:Yes, this is redundant. We removed the second sentence 'The average of all..'**

*P.20. I do not understand the newly added paragraph starting with The role of mass entrainment. In this section, only changes in snow mass are considered, while snow temperature and LWC are held constant. Hence, why do the authors point that the permutations did not include dry, cold snowcovers? Furthermore, the conclusion that snow quality (temperature, moisture) is more important than the snow amount, besides being not supported by the presented data, appears in complete contradiction with what is said in the following section. Please clarify this issue. More generally, I still believe that discussing, if possible, the relative contributions of changes in the initial mass versus changes in the entrained mass would be interesting.*

**ANSWER: Yes, it is in contradiction. We removed this paragraph. We were refereeing to entrainment in general, considering both dry and wet flows. This has no place in the present analysis which refers only to wet flows.**

*Sec.4.3. In their answer to J.T. Fischer, the authors mention that this section dealing with grid resolution will be amended to make its message clearer. Yet, this section was in fact only marginally modified in the revision. Similarly, my query to expand the last paragraph of the section, was not really addressed (this last paragraph has actually been removed). I think that expanding on the answer to J.T. Fischer, and adding a paragraph comparing the influence of changes in resolution versus changes in initial and boundary conditions, would be a great addition in the context of the present study, compared to a mere naked investigation of the effect of resolution.*

[revised manuscript text omitted]
 (%) upper | LWC (%) lower | Erosion height (m) upper | Erosion height (m) lower | Erosion height gradient (m/100m) upper | Erosion height gradient (m/100m) lower | density (kg/m$^3$) upper | density (kg/m$^3$) lower | volwater (mm/m) upper | volwater (mm/m) lower | temperature (°C) upper | temperature (°C) lower | temperature gradient (°C/100m) upper | temperature gradient (°C/100m) lower | erodibility (-) upper | erodibility (-) lower |
|---|---|---|---|---|---|---|---|---|---|---|---|---|---|---|---|---|
| Gruenbodeli | 1.45 | - | 0.56 | 0.00 | 0.02 | - | 197 | - | 8.1 | - | -0.2 | - | 0.0 | - | 0.8 | - |
| Salezer | 1.89 | - | 0.95 | 0.00 | 0.03 | - | 317 | - | 18.0 | - | 0.0 | - | 0.0 | - | 0.7 | - |
| Gatschiefer | 0.00 | 1.47 | 0.55 | 0.95 | 0.03 | 0.04 | 185 | 360 | 0.0 | 14.0 | -1.0 | 0.0 | 0.0 | 0.0 | 0.6 | 0.7 |
| Braemabuhl 2013 | 2.97 | - | 1.11 | 0.00 | 0.04 | - | 353 | - | 33.0 | - | 0.0 | - | 0.0 | - | 0.6 | - |
| Drusatscha | 3.41 | - | 0.54 | 0.00 | 0.02 | - | 291 | - | 18.4 | - | 0.0 | - | 0.0 | - | 0.6 | - |
| MO-4 Andina Chile | 2.44 | - | 0.90 | 0.00 | 0.03 | - | 296 | - | 22.0 | - | 0.0 | - | 0.0 | - | 0.6 | - |
| Grengiols | 0.00 | 4.67 | 0.43 | 0.60 | 0.03 | 0.00 | 175 | 270 | 0.0 | 28.0 | -7.4 | 0.0 | 1.5 | 0.0 | 0.7 | 0.8 |
| Verbier Mont Rogneux | 3.00 | - | 0.60 | 0.00 | 0.02 | - | 317 | - | 18.0 | - | 0.0 | - | 0.0 | - | 0.6 | - |
| Verbier Ba Combe | 2.59 | - | 0.58 | 0.00 | 0.02 | - | 349 | - | 15.0 | - | 0.0 | - | 0.0 | - | 0.6 | - |
| Braemabuhl verbauung | 0.00 | 1.41 | 0.25 | 0.85 | 0.00 | 0.04 | 158 | 335 | 0.0 | 12.0 | -2.0 | 0.0 | 0.0 | 0.0 | 0.8 | 0.8 |
| Braemabuhl Wildi | 0.00 | 1.25 | 0.30 | 0.80 | 0.00 | 0.03 | 164 | 335 | 0.0 | 10.0 | -2.0 | 0.0 | 0.0 | 0.0 | 0.6 | 0.6 |
| CV-1 Andina Chile | 1.51 | - | 0.37 | 0.00 | 0.00 | - | 359 | - | 5.6 | - | -0.1 | - | 0.0 | - | 0.6 | - |

[revised manuscript text omitted]